



# Fluorescence spectra of atmospheric aerosols

Jens Reichardt[1], Felix Lauermann[1], and Oliver Behrendt[1]

[1]Deutscher Wetterdienst (DWD), Meteorologisches Observatorium Lindenberg, Germany

**Correspondence:** Jens Reichardt (jens.reichardt@dwd.de)

**Abstract.** This study summarizes the results of 14 years of fluorescence measurements with the spectrometric fluorescence and Raman lidar RAMSES at the Lindenberg Meteorological Observatory, Germany. The focus is on findings that can only be obtained by spectrometric measurements and not with a single discrete fluorescence receiver channel. The measurement parameters that are useful in the spectral analysis are introduced. In particular, this includes the spectrum of the fluorescence

capacity, which has proven to be instructive. A new method is described that has been employed to determine the aerosol source regions. It is based on the combination of ensemble back-trajectory calculations with satellite measurements of fires and aerosol plumes. A total of 14 RAMSES measurements are discussed, which represent a selection of the most interesting measurement nights in the years 2020 to 2023. The emphasis is on biomass burning aerosol (BBA) in the free troposphere, but measurements of Saharan dust or boundary-layer aerosol are also provided. Excited at a wavelength of 355 nm, the BBA

fluorescence spectrum has a rounded shape with a maximum between about 500 and 550 nm. With height, it becomes increasingly Gaussian, and a shift towards longer wavelengths is generally observed (red shift). However, BBA layers which exhibit an opposite dependence (blue shift) can be found in specific cases as well. Overall, the spectral fluorescence capacity is high with values up to over $9 \times 10^{-6}$ nm$^{-1}$. Fluorescence spectra of Saharan dust, on the other hand, are skewed to short wavelengths. The fluorescence maxima are below 500 nm, and a linear decrease in the spectral backscattering coefficient can

be seen at longer wavelengths; the spectral fluorescence capacity is low ($< 1 \times 10^{-6}$ nm$^{-1}$). Our statistical analyses show that the correlations between the characteristics of the fluorescence spectra and other parameters are relatively weak. Of the atmospheric state variables, the ambient temperature correlates best, and of the elastic-optical particle properties it is the particle depolarization ratio. In addition, indications are found for both BBA and Saharan dust that the spectral shape is influenced not only by the type but also by the source region of the aerosols, which would allow a more accurate aerosol typing if further

substantiated. The importance of spectral fluorescence measurements for investigations of aerosol-cloud interaction is also highlighted. Measurement examples reveal cirrus nucleation at low supersaturation on contact with an optically extremely thin BBA filament at the tropopause, and provide the clearest indication to date of interaction, rather than coexistence, between clouds with supercooled water droplets and BBA using lidar. This groundbreaking study opens up a new field in atmospheric and aerosol research with exciting prospects for future studies, some of which will be presented.



## 1 Introduction

Spectrometric studies with lidar have been carried out for a long time. Initially limited to the investigation of the fluorescence of hard targets, such as the monitoring of oceans for oil pollution (Sato et al., 1978) or the assessment of biodegradation of historic building facades (Raimondi et al., 1998), the first attempts at application in atmospheric research were made in the early 2000s. However, these were at first restricted to the investigation of the Raman spectrum of liquid water to derive the water content of clouds in the boundary layer and remained qualitative (Arshinov et al., 2002; Kim et al., 2009; Sakai et al., 2012). The year 2012 then turned out to be pivotal for spectrometric lidar measurements. Sugimoto et al. (2012) expanded their scope to fluorescence spectra of aerosols, while Reichardt (2012) finally solved the problem of an absolute and robust calibration of spectral lidar measurements. First applied to Raman spectra of cirrus ice, the approach was later adopted for the calibration of aerosol fluorescence spectra as well. It is probably justified to say that these two achievements led to an increase in interest in lidar-based spectrometry of clouds and aerosols.

The present study is the essence of 14 years of fluorescence measurements with the spectrometric fluorescence and Raman lidar RAMSES at the Lindenberg Meteorological Observatory, Germany. Designed and operated since 2005 as a conventional high-performance multiparameter Raman lidar (Reichardt et al., 2012), RAMSES was gradually extended to its spectral measurement capabilities starting in 2010 by integrating a total of three spectrometers. The so-called water spectrometer was the first to be commissioned in 2011 for measurements of the Raman spectra of water in its three phases and aerosol fluorescence in the short-wave VIS range (Reichardt, 2014), followed by the VIS and UVA spectrometers optimized for fluorescence measurements in 2015 and 2018 (Reichardt et al., 2023a).

The unique features of RAMSES allowed Reichardt (2014) to discuss absolutely calibrated cloud Raman and aerosol fluorescence spectra in detail for the first time. The now widely used intensive parameter of spectral fluorescence capacity was introduced (which is similar to the lidar fluorescence ratio of Sugimoto et al., 2012) and its scientific value for aerosol typing was demonstrated. Reichardt et al. (2018) were then the first to show that fluorescence can be utilized to peer into clouds and reveal the coexistence of aerosols, which is of great benefit in the research of cloud formation. Furthermore, investigations were published that analyzed the negative effect of aerosol fluorescence on the measurements of the water content of clouds (Reichardt et al., 2022) or the water vapor mixing ratio (Reichardt et al., 2023a) and described methods for its correction.

Probably inspired by the early spectral aerosol fluorescence studies, fluorescence lidars have subsequently been developed that employ a single or a small number of broadband discrete fluorescence detection channels instead of a spectrometer (e.g., Rao et al., 2018; Veselovskii et al., 2020; Hu et al., 2022; Veselovskii et al., 2023; Gast et al., 2024). These experimentally simpler instruments can certainly make a contribution to aerosol research, e.g. in aerosol typing or in the coexistence of clouds and aerosols, but sophisticated spectrometric instrumentation is required for investigations focusing on aerosol fluorescence itself or on aerosol-cloud interaction. Only a few instruments are equipped accordingly (e.g. the LITES testbed of Tatarov and Müller, 2021), but RAMSES is, to the authors' knowledge, the only one currently in routine operation. This study presents the latest findings in these research areas.



Section 2 provides a brief description of the RAMSES instrument and defines the measurement parameters that are useful for analyzing the aerosol fluorescence spectra. Furthermore, the approach developed to determine the aerosol source region is explained in detail and an overview of the measurements featured in this study is given. Section 3 then presents and discusses the results of four years of aerosol fluorescence measurements (2020–2023). In the authors' opinion, such an extensive data set was necessary to do justice to the complexity of the problem and to avoid premature conclusions. Section 3.1 first presents the basic characteristics of the fluorescence spectra of the two aerosol types that make up the bulk of the fluorescence measurements over Lindenberg according to the current state of research. Specifically, these are BBA (Sect. 3.1.1) and mineral dust (Sect. 3.1.2). This is followed by a discussion of the dynamics and variability of the BBA fluorescence spectrum (Sect. 3.2). It turns out that there is no simple relationship between the BBA fluorescence spectrum observed in a particular measurement and the atmospheric state, the elastic particle properties, or the aerosol history. For a better understanding, statistical analyses are required, which are shown in Sect. 3.3. Next, Sect. 3.4 reports measurement results on the interaction of BBA with clouds, which underline the advantages of spectral fluorescence measurements in cloud research. The effect of water uptake by aerosol particles in the boundary layer on their fluorescence properties is also discussed. Finally, Sect. 4 summarizes the main findings of our investigation and develops perspectives for future studies.

## 2 Instrument and methods

### 2.1 Instrument

RAMSES (Raman lidar for moisture sensing) is the high-performance spectrometric fluorescence and Raman lidar of the German Meteorological Service (DWD). Located at the Lindenberg Meteorological Observatory east-southeast of Berlin, Germany, it has been in routine operation since 2005. The measurement parameter set of the instrument has been extended over time by upgrading the receiver system, the current experimental configuration was completed in 2018. The RAMSES setup and the spectral measurement methods have already been documented in detail (Reichardt et al., 2012; Reichardt, 2014; Reichardt et al., 2023a). Therefore, only a brief description is given here, as it is necessary for understanding the aerosol measurements.

The RAMSES transmitter emits the UV light pulses (354.7 nm) of a 30-Hz injection-seeded frequency-tripled Nd:YAG laser; average UV output power is about 15 W. Fundamental and second-harmonic generation light is effectively suppressed by a combination of dichroic beam splitters and a Pellin Broca prism, which is a prerequisite for the accurate measurement of fluorescence spectra. The receiver has two separate branches. The near-range receiver consists of a Newtonian telescope (300 mm diameter), which is connected via a fiber to a polychromator with three discrete receiver channels and the so-called UVA spectrometer (378–458 nm spectral range). In the far-range receiver, a Nasmyth-Cassegrain telescope (790 mm diameter) and the polychromator are directly coupled, i.e. without a fiber. There are 9 discrete receiver channels and two spectrometers, the water spectrometer (385–410 nm spectral range; measurement of water in all three phases) and the VIS spectrometer (440–750 nm effective spectral range). Combination of signals of the discrete receiver channels allows the measurement of the elastic particle properties, specifically the particle extinction coefficient ($\alpha_{par}$), particle backscatter coefficient ($\beta_{par}$) and thus particle





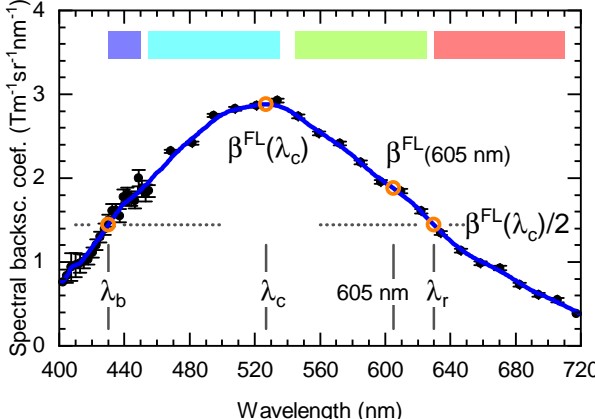

**Figure 1.** Aerosol fluorescence spectrum as measured with RAMSES at 4.8 km around 01:15 UTC on 27 May 2023. Calibrated data from the UVA spectrometer and the VIS spectrometer are merged (black symbols with statistical-error bars), interpolated and smoothed with a boxcar average of 25 nm width (blue curve). Wavelengths of the maximum of the fluorescence spectrum ($\lambda_c$), of the spectrum half width on the blue shoulder ($\lambda_b$) and on the red shoulder ($\lambda_r$) and of 605 nm, and the corresponding spectral fluorescence backscatter coefficients ($\beta^{FL}$, orange symbols) are visualized which are used as, or for the determination of, parameters that describe the aerosol spectral properties. Pastel blue, cyan, green, and red color bars show spectral integration ranges (430–450, 455–535, 545–625 and 630–710 nm, respectively) of false color parameters of the aerosol fluorescence spectrum. To obtain the spectrum, 1200 s of lidar data have been integrated. The resolution of the raw data is 60 m, signal profiles have been smoothed with a sliding-average length of 7 height bins. The entire measurement night of 26–27 May 2023 is presented in Fig. 10.

lidar ratio ($S_{par}$), and particle depolarization ratio ($\delta_{par}$), as well as of temperature ($T$) and the water vapor mixing ratio. The water vapor mixing ratio can also be determined with the UVA spectrometer, which has the advantage that it can be corrected for fluorescence, however, the measurement range is slightly reduced as a result. Relative humidity is calculated using either temperature data from 6-hourly radiosonde launches on site or from RAMSES, measurement conditions permitting. Because

95  aerosols are the subject of the investigations and not clouds, $\alpha_{par}$ and $S_{par}$ were not corrected for multiple scattering.

## 2.2 Fluorescence spectra: analysis and presentation

Fluorescence spectra presented in this study have been obtained by merging the spectral data of the RAMSES UVA and VIS spectrometers following the procedure described in detail by Reichardt et al. (2023a). An example is given in Fig. 1. The measured spectral fluorescence backscatter coefficients ($\beta^{FL}$) are visualized as black symbols, bars indicate statistical errors.

100  At shorter wavelengths, the data originate from the UVA spectrometer; due to the smaller detector bandwidth, the data points are more closely spaced and have a statistical error approximately five times higher than the VIS data > 460 nm due to the smaller bandwidth-telescope diameter product. For spectral analysis, the spectra are first interpolated to a uniform wavelength vector with a step size of 1 nm and then smoothed with a boxcar average of 25 nm width (blue curve).





In order to be able to visualize the development of the aerosol spectra in time and space, several parameters were defined
(Fig. 1). Specifically, these are properties of the spectrum shape:

(a) Wavelength of the maximum of the fluorescence spectrum ($\lambda_{\mathrm{c}}$);

(b) Wavelength of the spectrum half width on the short-wavelength (blue) shoulder ($\lambda_{\mathrm{b}}$) defined as $\beta^{\mathrm{FL}}(\lambda_{\mathrm{b}}) = 0.5\beta^{\mathrm{FL}}(\lambda_{\mathrm{c}})$
with $\lambda_{\mathrm{b}} < \lambda_{\mathrm{c}}$. Note that $\lambda_{\mathrm{b}}$ is not always provided, in particular when the aerosol spectra exhibit a relatively short center
wavelength (CWL) or UVA spectrometer data are not available;

(c) Wavelength of the spectrum half width on the long-wavelength (red) shoulder ($\lambda_{\mathrm{r}}$) defined as $\beta^{\mathrm{FL}}(\lambda_{\mathrm{r}}) = 0.5\beta^{\mathrm{FL}}(\lambda_{\mathrm{c}})$
with $\lambda_{\mathrm{r}} > \lambda_{\mathrm{c}}$; and

(d) Ratio of the spectral fluorescence backscatter coefficients measured at $\lambda_{\mathrm{c}}$ and 605 nm, defined as
$R_{605}^{\mathrm{CWL}} = \beta^{\mathrm{FL}}(\lambda_{\mathrm{c}})/\beta^{\mathrm{FL}}(605\ \mathrm{nm})$. $R_{605}^{\mathrm{CWL}}$ can be regarded as a measure of the skewness of the fluorescence spectrum.

Furthermore, properties of the spectrum are specified in the range of the four spectral bands 430–450, 455–535, 545–625,
and 630–710 nm, to which the blue, cyan, green, and red (false) colors, respectively, are assigned. The violet spectral band
(395–407 nm), which was of importance in our preceding study (Reichardt et al., 2023a), is not considered in the present work.
The false color properties are:

(e) Fluorescence backscatter coefficient ($\mathcal{B}_{\mathrm{color}}^{\mathrm{FL}}$) defined as the integral of $\beta^{\mathrm{FL}}$ over the specific wavelength range;

(f) Mean spectral fluorescence backscatter coefficient ($\beta_{\mathrm{color}}^{\mathrm{FL}}$) defined as the quotient of $\mathcal{B}_{\mathrm{color}}^{\mathrm{FL}}$ and the spectral width of the
specific wavelength range; and

(g) Mean spectral fluorescence capacity ($C_{\mathrm{color}}^{\mathrm{FL}}$) defined as $C_{\mathrm{color}}^{\mathrm{FL}} = \beta_{\mathrm{color}}^{\mathrm{FL}}/\beta_{\mathrm{par}}$.

Spectral fluorescence capacity, introduced as an aerosol layer-averaged quantity by Reichardt (2014) but subsequently pro-
vided as profiles (Reichardt et al., 2018), relates elastic and inelastic scattering properties of aerosol particles and quantifies
how strongly a certain type of aerosol may fluoresce upon UV excitation (at a wavelength of 355 nm in the case of RAMSES),
if $\beta_{\mathrm{par}}$ is viewed as an approximate measure of aerosol content. Finally, also discussed in this study is the

(h) Spectrum of fluorescence capacity defined as $\beta^{\mathrm{FL}}(\lambda)/\beta_{\mathrm{par}}$.

Previously, we experimented with other forms of representation to better understand the dynamics of the aerosol spectrum
(e.g., normalized to the maximum of the fluorescence spectrum or to a specific wavelength range, Reichardt et al., 2023b), but
all had disadvantages. In the course of our investigations, the $\beta^{\mathrm{FL}}/\beta_{\mathrm{par}}$ spectra proved to be the most instructive.





### 2.3 Trajectory analysis

#### 2.3.1 Approach

To analyze the origin of the observed aerosol layers, we performed a trajectory analysis with the Hybrid Single-Particle La-grangian Integrated Trajectory (HYSPLIT) model (Stein et al., 2015; Rolph et al., 2017). An ensemble of 27 trajectories was back-calculated over a 315-hour time span. To determine the source region for the BBA cases, additionally fire data from the Moderate Resolution Imaging Spectroradiometer (MODIS) was included in the analysis, using NASA's Fire Information for Resource Management System (FIRMS). In order to better estimate the strength of the fires and thus the potential with regard to the amount of smoke emitted, the individual MODIS fire pixels were aggregated into larger fire regions and the fire radiative power (FRP) of the individual fire pixels was accumulated. Fire regions with a FRP of less than $500 \ \mathrm{Wm^{-1}}$ were excluded from further consideration, as sample analyses showed no to little smoke emission from these sources.

Figure 2 shows as an example the 27 back-trajectories of the HYSPLIT ensemble for two BBA layers measured with RAMSES in the nights of 11–12 September 2020 and 8–9 August 2021. In addition, the fire regions are highlighted with colored circles. Only trajectories that pass close to fires are colored, all other trajectories are shown in gray. To account for the increasing uncertainty of the trajectory calculation with back-calculation time, trajectories were classified as close to fires if they passed a fire within a maximum distance of one kilometer times the duration of the back calculation (e.g., for a back-calculation time span of $100 \ \mathrm{h}$, the maximum distance allowed was $100 \ \mathrm{km}$). The two analyses differ in the number of possible source regions for the observed aerosol layer. In Fig. 2a, only the strong fires in the Western United States are crossed by the trajectories while, in contrast, in Fig. 2b several trajectories each pass over both the fires in Central and Western Canada as well as the fires in Russia. The example demonstrates that the combination of HYSPLIT trajectories and FIRMS fire data is sufficient for identifying the source region of the BBA in cases with only one potential fire region, whereas for more complex fire situations, and for a more precise determination of the aerosol source and transport time, additional data are required.

In our study, we used the daily global UV aerosol index data from the Suomi National Polar-orbiting Partnership (NPP) Ozone Mapping and Profiling Suite (OMPS, McPeters, 2024) as supplementary information. Figure 3 presents the analysis for the RAMSES measurement at 01:00 UTC on 12 September 2020 as an illustrative example, which was selected because it provides an overview of all situations that might be encountered. In this display, HYSPLIT, FIRMS, and OMPS data are combined. For all 27 ensemble members (Fig. 2a), the aerosol index is plotted for the geographical coordinates of the air parcel along the back-trajectory at the corresponding time step (horizontal bars). For an optimized visualization, values of aerosol index were set to $-2$, $0$, and $10$ if missing, $< 0$, and $> 10$, respectively. Depending on the number and intensity of the fires crossed and the corresponding behavior of the aerosol index, different scenarios can be distinguished:

S1: If no fires were detected close to the trajectory and the aerosol index value is low (e.g., trajectory 03 before 9 September), the trajectory most likely did not cross any fire region.

S2: If no fires were detected close to the trajectory, but the aerosol index value is high (e.g., trajectory 14 after 6 September), either a fire was crossed by the trajectory but not detected by the satellite, or an area with a smoke plume instead of a





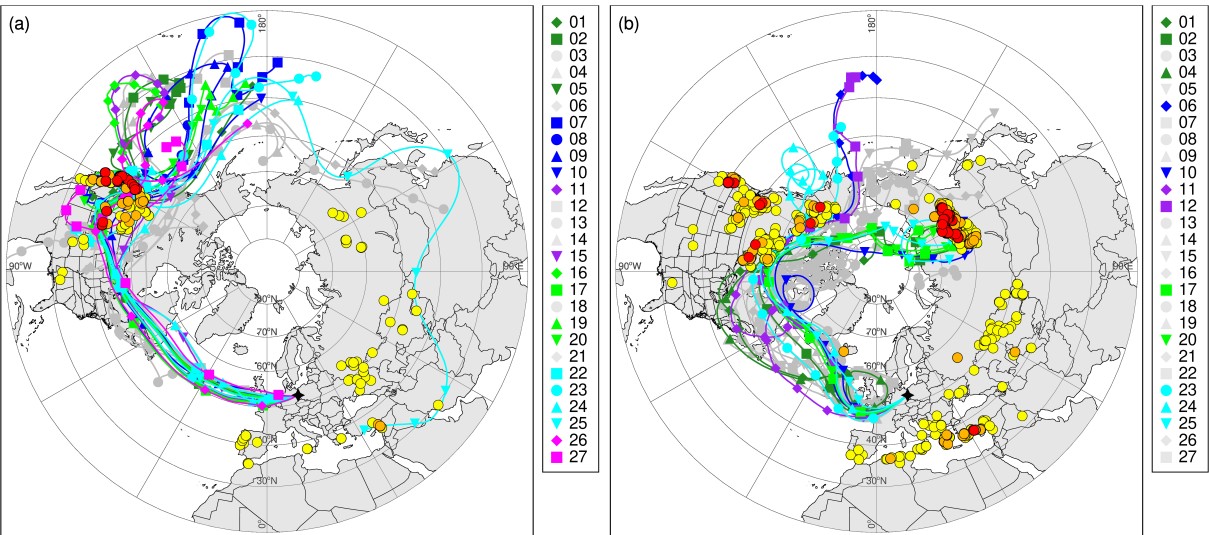

**Figure 2.** Two examples of the HYSPLIT back-trajectory ensemble for the starting points **(a)** 12 September 2020, 01:00 UTC, 5.0 km altitude, and **(b)** 9 August 2021, 23:00 UTC, 5.4 km altitude. Trajectories close to fires are indicated by colored lines and symbols, all other trajectories of the ensemble are shown in gray. Fires are represented by filled circles and color-coded according to the accumulated fire radiative power (FRP) of a fire region; yellow, orange, and red circles indicate FRP values between 500 and 5000, 5000 and 25000, and $> 25000\ \mathrm{Wm^{-2}}$, respectively.

fire was crossed, with the trajectory probably be located at a height other than that of the smoke plume (as OMPS only provides column integrated values, a trajectory can be located above or below a smoke layer).

S3: If fires were detected close to the trajectory, but the aerosol index value is low (e.g. trajectory 15), the trajectory most likely did either cross a weak fire with less smoke emission or the detected fire was only a hot spot without any smoke emission.

S4: If fires were detected close to the trajectory, but the aerosol index value is only temporarily enhanced after the trajectory crosses the fires (e.g., trajectory 24), the trajectory most likely did not cross the fire within the altitude range of the smoke
layer, but above it.

S5: If fires were detected close to the trajectory, the aerosol index value rises as the trajectory crosses the fires and henceforth shows consistently elevated values (e.g., trajectory 20), then the trajectory most likely did cross a fire region and the height of the trajectory is within the vertical range of the aerosol plume.

Only those trajectories of the ensemble matching scenarios S3 or S5 were selected for further analysis, with scenario S5
having the highest probability that the trajectory adequately represents the transport path of the observed aerosol. For the 13 BBA cases analyzed, a total of 116 ensembles were calculated for different start times and heights. In 56, 35, and 15 of these ensembles, the number of representative trajectories was between 1 and 9, 10 and 18, and 19 and 27, respectively. No





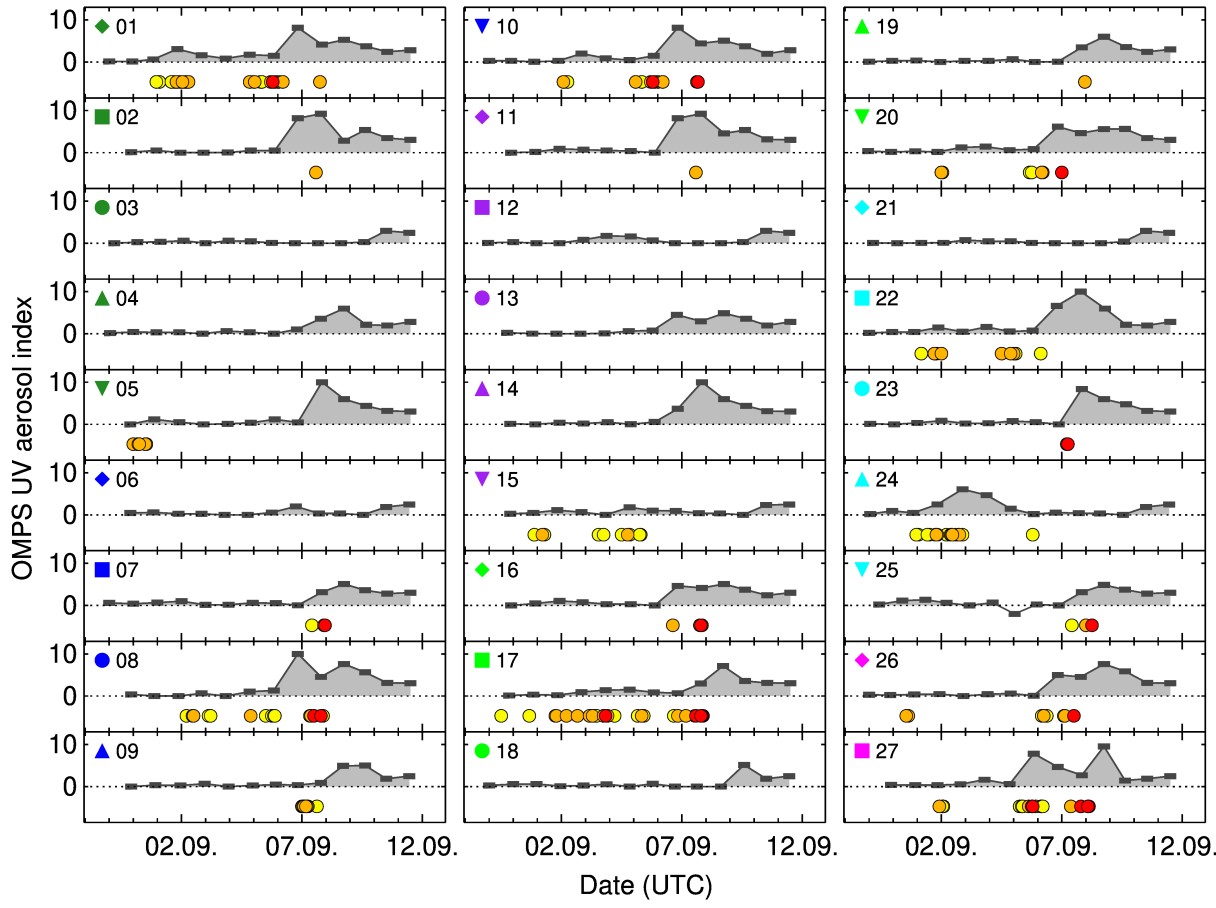

**Figure 3.** OMPS aerosol index values along the path of the 27 ensemble trajectories (horizontal gray bars) for the starting point 12 September 2020, 01:00 UTC, 5 km (Fig. 2a). Fires close to trajectories are indicated by filled circles, the colors match those in Fig. 2.

representative trajectory was found in 10 ensembles. As a final step in our analysis, we calculated additional aerosol transport parameters, such as the advection time and the mean temperature along the path after passing a fire region, as mean values over
all plausible trajectories using the FIRMS fire data and the HYSPLIT meteorological data.

Similar investigations on BBA events were carried out by Hu et al. (2022) and Ortiz-Amezcua et al. (2017), for example. Among other data, Hu et al. (2022) also used HYSPLIT back-trajectories and OMPS aerosol index data. However, only individual trajectories were calculated instead of an ensemble, and no information on the exact location of the fires was exploited, so that just a rough delimitation of the source region was possible. Ortiz-Amezcua et al. (2017) utilized both HYSPLIT en-
semble runs and MODIS fire data, but apparently the link between the fire regions and the path of the trajectories was loosely established at best. In addition, Cloud-Aerosol Lidar and Infrared Pathfinder Satellite Observation (CALIPSO) data were evaluated to obtain information on the spatial distribution of the aerosol. This has the advantage over OMPS that the height of the





aerosol layer can also be determined, but the spatial coverage is more limited, which makes it more difficult to track aerosol layers. As a result, Ortiz-Amezcua et al. (2017) were likewise only able to provide a broad delineation of the BBA origin.

For the analysis of Saharan dust events, the height of the trajectories above the ground was used to find the source region. A region was identified as a possible dust source if, firstly, a trajectory crossed this region at a height of less than 300 m above the ground, secondly, the OMPS aerosol index increased during the low overpass and, thirdly, it maintained elevated values thereafter. Sensitivity studies revealed a satisfactory correlation between RAMSES observations of Saharan dust and back-trajectories with low Sahara overpass heights and high aerosol concentrations. As demonstrated by Middleton and Goudie
(2001) and Baddock et al. (2009), the method of analysis could be refined by incorporating satellite data and knowledge of typical emission areas for Saharan dust. However, as only one dust case was included in this study, the authors considered it sufficient to exclude fires as the aerosol source and to roughly localize the dust source.

### 2.3.2 Overview of measurements

The trajectory analyses are time-consuming and were therefore only carried out for a small selection of what are now hundreds
of measurement nights with fluorescing aerosols. The selected cases had the best data quality and were considered the most intriguing. Of these, those for which the source region could only be determined with insufficient certainty were then also discarded. Table 1 gives an overview of the measurements covered in this publication. A total of 14 aerosol events are discussed, for each date a general characterization is provided. This includes the dominant type of aerosol (in the free troposphere; no attempt was made to assess the source regions of the boundary-layer aerosols), the temporal and spatial evolution of the
fluorescence features, and the presence of clouds. The emphasis is on BBA (13 cases), but a measurement with Saharan dust (22–23 February 2022) is also included to illustrate the dependence of the spectral properties on the aerosol type. Aerosol fluorescence is referred to as homogeneous, layered, or dynamic if the spectral properties varied little with height and overnight, changed only with height, or were highly variable in space and time, respectively. For better guidance, the relevant figure numbers are listed for each measurement night; the number given first is always for the overview display. For the presentation
of the two measurement cases in August 2021, reference is made to earlier publications.

  In the overview plots, patterns emerge in the fluorescence properties that indicate aerosol particles with the same inelastic properties. They form layers that often persist for several hours and show little height variability. For these layers the source regions were determined. In addition to the time and height interval of their occurrence over Lindenberg, only the results of the trajectory analysis are summarized in Tab. 1 which are used in the later discussion. Specifically, these are the origin, the
transport duration and the average temperature along the trajectory. It can be seen that the BBA originated predominantly from forest fires on the west coast of North America, particularly in 2020 and 2021. Other sources also contributed, but with less frequency. These include Siberia, the Iberian Peninsula and Eastern Canada. The Saharan dust originated mainly from the deserts of Algeria, but there have always been contributions from other western or central regions of North Africa as well. Transport time and mean temperature are therefore vague and have not been listed.




**Table 1.** Overview of the measurement nights presented in this study, and results of the back-trajectory analysis for selected aerosol layers.

| Date[1] | Time | Height | Origin | Conf.[2] | $t_{\mathrm{mean}}$ | $\overline{T}_{\mathrm{mean}}$ | Characterization | Figures |
|---|---|---|---|---|---|---|---|---|
| (yy/mm/dd) | (UTC) | (km) | | | (h) | (°C) | | |
| 20/09/11 | 22:45–00:40 | 4.0–5.0 | WUS (4–7) | 5 | 130 | −6 | BBA, dynamic | A1 |
| | 00:00–04:00 | 6.0–8.0 | WUS (6–7) | | 125 | −20 | | |
| | 01:00–04:00 | 4.4–4.6 | WUS (4) | | 120 | −4 | | |
| | | 4.9–5.1 | WUS (6-7) | | 120 | −9 | | |
| 20/09/13 | 18:00–04:00 | 3.0–3.8 | WUS (9) | 2 | 235 | 1 | BBA, layered, | A2, 17 |
| | 19:00–01:00 | 6.5–8.0 | WUS (9), WCA | | 190 | −25 | fil. (14 km, Ci)[3] | |
| | 22:00–04:00 | 4.1–4.5 | WUS (7–8), WCA | | 200 | −12 | | |
| 20/09/19 | 20:30–00:00 | 7.5–8.5 | WUS (4) | 1 | 260 | −22 | BBA, layered, | A3 |
| | 20:30–03:00 | 9.0–11.0 | CA, OR, WA | | 250 | −25 | Ci | |
| 21/02/22 | 22:00–00:00 | 3.0–4.0 | WNA (7) | 1 | – | – | SD, dynamic, | 7–9 |
| | 01:00–05:00 | 3.0–4.0 | CNA (4) | | – | – | Ac | |
| 21/08/09 | 00:00–01:00 | 3.5–4.5 | WCA | 2 | 200 | 0 | BBA, layered, | Reichardt et. |
| | | 5.0–5.5 | ERU | | 250 | −12 | fil. (11 km, Ci) | al. (2023b) |
| 21/08/21 | 01:00–02:00 | 6.0–6.9 | WCA, WUS (3) | 4 | 130 | −16 | BBA, dynamic, | Reichardt et. |
| | | 7.0–10.0 | WCA, WUS (2–5) | | 120 | −24 | Ci, Ac | al. (2023a) |
| 22/07/19 | 21:00–02:00 | 3.0–4.0 | PT, ES, MA | 4 | 65 | 6 | BBA, SD, layered | A4 |
| | | 4.2–4.7 | PT, ES | | 60 | 0 | | |
| | | 5.2–5.8 | PT, ES | | 55 | −12 | | |
| | | 6.2–6.8 | PT, ES | | 65 | −20 | | |
| 22/07/20 | 20:00–01:00 | 3.5–4.5 | PT, ES, MA | 3 | 115 | −5 | BBA, SD, layered | A5 |
| | | 5.1–5.6 | PT, ES | | 105 | −13 | | |
| | | 6.1–6.8 | PT, ES | | 85 | −17 | | |
| 23/05/26 | 21:30–01:00 | 6.1–7.0 | WCA | 3 | 135 | −30 | BBA, dynamic | 10–12 |
| | 22:00–02:00 | 10.8–12.0 | WCA | | 135 | −53 | | |
| | 00:00–01:00 | 4.0–6.0 | WCA | | 140 | −23 | | |
| | 00:00–02:00 | 8.2–9.8 | WCA | | 95 | −44 | | |
| 23/06/01 | 21:45–22:30 | 3.9–4.4 | AB | 3 | 120 | −10 | BBA, dynamic | A6 |
| | 00:00–01:15 | 5.2–6.0 | AB, ECA | | 130 | −27 | | |
| | 00:30–01:30 | 7.5–10.0 | AB | | 115 | −50 | | |





*continued from previous page*

| Date | Time | Height | Origin | Conf. | $t_{\mathrm{mean}}$ | $\overline{T}_{\mathrm{mean}}$ | Characterization | Figures |
|---|---|---|---|---|---|---|---|---|
| (yy/mm/dd) | (UTC) | (km) | | | (h) | (°C) | | |
| 23/06/04 | 21:00–01:00 | 3.9–5.3 | SK | 2 | 155 | −17 | BBA, layered | A7 |
| | | 5.8–7.0 | SK | | 115 | −33 | | |
| | | 8.0–8.5 | SK, ECA | | 120 | −38 | | |
| 23/07/04 | 21:00–23:00 | 4.0–6.0 | ECA, WCA | 4 | 100 | −13 | BBA, homogeneous, | 4–6, |
| | 01:00–02:00 | 3.2–5.5 | ECA, WCA | | 115 | −12 | Ac | 19, 20 |
| 23/07/09 | 20:30–22:30 | 8.0–10.0 | ECA | 3 | 100 | −40 | BBA, homogeneous, | A8, |
| | 20:30–00:00 | 5.1–7.0 | ECA | | 125 | −19 | Ci, Ac | 21, 22 |
| 23/09/30 | 17:30–23:00 | 4.0–6.0 | WCA | 5 | 180 | −26 | BBA, homogeneous, | A9 |
| | 00:00–04:00 | 2.0–4.0 | WCA | | 190 | −13 | Ci, As, Ac | |

[1] The afternoon date of the measurement night is provided.

[2] Conf. – confidence in the determination of origin on a scale from 1 to 5, with 5 being the highest level.

[3] fil. – filament of BBA aerosol. Center height is indicated, cirrus formed at its lower edge.

$t_{\mathrm{mean}}$ – ensemble-averaged transport time, $\overline{T}_{\mathrm{mean}}$ – ensemble-averaged mean temperature.

AB – Alberta, CA – California, ES – Spain, MA – Morocco, OR – Oregon, PT – Portugal, SK – Saskatchewan, and WA – Washington.

ECA – Eastern Canada, ERU – Eastern Russia, WCA – Western Canada, and WUS – Western United States (number of states in parentheses).

CNA – Central North Africa, and WNA – Western North Africa (number of countries in parentheses).

BBA – biomass burning aerosol, SD – Saharan dust, Ci – cirrus, Ac – altocumulus, and As – altostratus.

## 3 Results and discussion

The results of four years of aerosol fluorescence measurements (2020 to 2023) are presented and discussed. Section 3.1 first describes the basic properties of the fluorescence spectra of the two most frequently observed aerosol types (BBA and mineral dust). A discussion of the dynamics and variability of the BBA fluorescence spectrum follows in Sect. 3.2. Statistical analyses were performed to investigate the relationship between the BBA fluorescence spectrum and atmospheric variables, the elastic particle properties or the aerosol history; the results are shown in Sect. 3.3. Finally, Sect. 3.4 highlights measurements of the

interaction of BBA with clouds. The effect of water uptake by aerosol particles in the boundary layer on their fluorescence properties is also addressed.

### 3.1 Fluorescence spectra of different aerosol types: general properties

Most aerosol events over Lindenberg in the free troposphere are associated with BBA and mineral dust. Other aerosol types are very rare. Volcanic aerosol, for example, was only measured once, namely after the eruption of Cumbre Vieja, Canary Islands,

in autumn 2021 (Hedelt et al., 2024), and pollen has not been measured at all in recent years. We therefore focus our studies on the former.



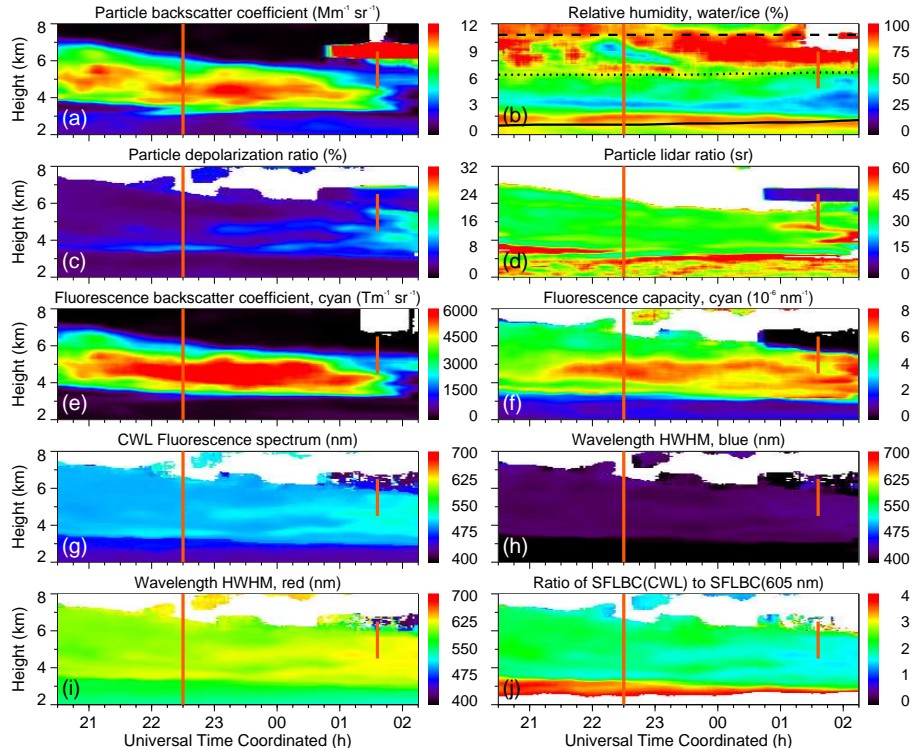

**Figure 4.** RAMSES measurement in the night of 4–5 July 2023 between 20:30 and 02:15 UTC. Temporal evolution of **(a)** particle backscatter coefficient, **(b)** relative humidity (with respect to water and ice above and below 0 °C, respectively; 0 °C, -15 °C and -30 °C isotherms indicated by black lines), **(c)** particle depolarization ratio, **(d)** particle lidar ratio (not corrected for multiple scattering), **(e)** fluorescence backscatter coefficient (cyan false color: spectrum integrated from 455 to 535 nm), **(f)** spectral fluorescence capacity (mean value, 455–535 nm), **(g)** wavelength of the maximum of the fluorescence spectrum (CWL), **(h)** wavelength of the spectrum half width (HWHM) on the blue shoulder, **(i)** wavelength of the spectrum half width on the red shoulder, and **(j)** ratio of CWL to 605-nm spectral fluorescence backscatter coefficients (SFLBC). The measurement at 22:30 UTC (time marked by vertical orange line) is analyzed in Figs. 5 and 6, the measurement at 01:36 UTC is discussed in Sect. 3.4.2. For each profile, 1200 s of lidar data are integrated, the calculation step width is 120 s. The resolution of the raw data is 60 m, signal profiles are smoothed with a sliding-average length increasing with height. White areas indicate where data were rejected by the automated quality control process.

### 3.1.1 Biomass burning aerosol (BBA)

Figure 4 depicts the RAMSES measurement during the night of 4–5 July 2023, it is a prime example of a BBA event with homogeneous properties. As in all overview plots in this publication, the elastic particle properties ($\beta_{\mathrm{par}}$, $\delta_{\mathrm{par}}$, $S_{\mathrm{par}}$) and the relative humidity are shown in the two upper panels, while the inelastic properties of the aerosol fluorescence as defined in Sect. 2.2 are displayed in the three lower panels. Incidentally, this BBA event is also discussed by Gast et al. (2024) using data obtained with a lidar system that is similar in performance to RAMSES but is only equipped with a single discrete fluorescence




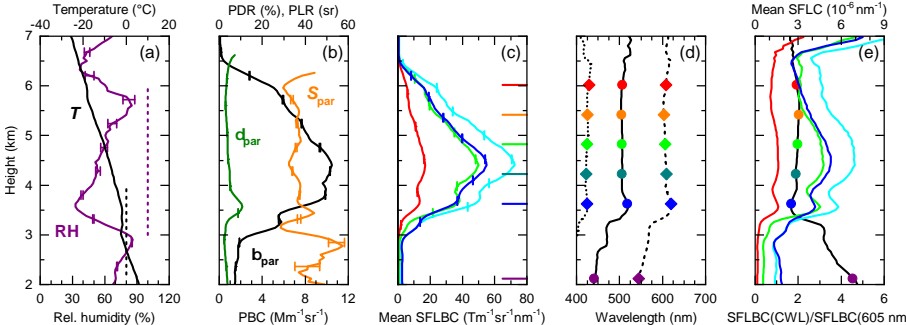

**Figure 5.** Measurement of the BBA layer at 22:30 UTC on 4 July 2023. Profiles of **(a)** relative humidity with respect to water or ice (RH; saturation level indicated by vertical dashed violet line) and temperature ($T$; freezing temperature indicated by vertical dashed black line), **(b)** particle backscatter coefficient (PBC, $\beta_{\mathrm{par}}$), particle depolarization ratio (PDR, $\delta_{\mathrm{par}}$) and particle lidar ratio (PLR, $S_{\mathrm{par}}$), **(c)** mean spectral fluorescence backscatter coefficients (SFLBC) for the spectral bands ranging from 430–450 nm (blue), 455–535 nm (cyan), 545–625 nm (green), and 630–710 nm (red), **(d)** wavelength of the maximum of the fluorescence spectrum (CWL, solid curve), wavelength of the spectrum half width on the blue shoulder (dotted curve) and wavelength of the spectrum half width on the red shoulder (dashed curve), and **(e)** ratio of CWL to 605-nm spectral fluorescence backscatter coefficients (black curve) and mean spectral fluorescence capacities (SFLC) for the blue, cyan, green, and red spectral bands (colored curves). Colored symbols and horizontal lines indicate altitudes for which the fluorescence spectra are presented in Fig. 6. Bars in panels **a–c** indicate statistical errors of the lidar measurement (omitted in panels **d** and **e** for the sake of conciseness). Temperature profiles were obtained by interpolating data from radiosondes launched every 6 hours on site.

detection channel. Our results demonstrate how much more information can be gained when a spectrometric fluorescence lidar is operated.

The BBA layer stretches from about 3 to 6 km in altitude; its origin was Eastern and Western Canada, transport time was about 96–144 h. The fluorescence backscatter coefficient in the cyan color spectrum ranges up to 6000 $\mathrm{Tm}^{-1}\mathrm{sr}^{-1}$, which are by far the highest $\mathcal{B}_{\mathrm{cyan}}^{\mathrm{FL}}$ values ever recorded over Lindenberg (Fig. 4e). The cyan mean spectral fluorescence capacity is also pronounced but not exceptional ($C_{\mathrm{cyan}}^{\mathrm{FL}}$ values up to $7 \times 10^{-6}$ nm$^{-1}$, Fig. 4f). In the plots of wavelengths $\lambda_{\mathrm{c}}$, $\lambda_{\mathrm{b}}$, and $\lambda_{\mathrm{r}}$, which characterize the shape of the spectrum, only weakly developed features can be seen until about 01:00 UTC; later, in line with

the reduction of $\mathcal{B}_{\mathrm{cyan}}^{\mathrm{FL}}$, they shift to higher values and resemble those of the filament at 3.6 km, which could be observed since the beginning of the measurement (Figs. 4g–4i). Similar patterns can be found in the particle number density-independent elastic parameters $\delta_{\mathrm{par}}$ and $S_{\mathrm{par}}$ (Figs. 4c and 4d), and in relative humidity, which is, however, anti-correlated (Fig. 4b).

The sudden change visible in all parameters slightly above 3 km indicates the transition from the boundary layer to the free troposphere. In the case of aerosol fluorescence, this change can be observed particularly well in the skewness $R_{605}^{\mathrm{CWL}}$ (Fig. 4j).

As we will see below, this is due to the completely different shapes of the fluorescence spectra of BBA and boundary-layer aerosol. Note that embedded in the humid layer above 6 km an altocumulus cloud entered the RAMSES field-of-view at 00:45 UTC. This incident will be investigated for a possible cloud effect on aerosol fluorescence in Sect. 3.4.2.




For a quantitative view, the profiles measured at 22:30 UTC are presented in Fig. 5 as an example. Within the BBA layer, the temperature is between $-20$ and $0\ °\mathrm{C}$, and relative humidity increases with altitude from $\sim 30\ \%$ to over $80\ \%$, and still the particle properties, both elastic and inelastic, are unaffected. Only around $3.6\ \mathrm{km}$, the center height of the aforementioned filament, differences are discernible. Particle depolarization ratio averages $4\ \%$, whereas lidar ratio ranges between $32\text{-}40\ \mathrm{sr}$. While such $\delta_{\mathrm{par}}$ values are not uncommon in our BBA data set, $S_{\mathrm{par}}$ values $< 40\ \mathrm{sr}$ are rare. In general, $S_{\mathrm{par}}$ behaves more erratically, especially in its relationship to the fluorescence properties, than $\delta_{\mathrm{par}}$, as our statistical analysis in Sect. 3.3.3 will show. In any case, at such low lidar ratios it can be ruled out that the BBA particles absorbed significantly.

The curves of the false-color mean spectral fluorescence backscatter coefficients (Fig. 5c) and capacities (Fig. 5e) exhibit nearly identical shapes in the BBA layer. This is not observed in the boundary layer, where $C_{\mathrm{blue}}^{\mathrm{FL}}$ falls below $C_{\mathrm{cyan}}^{\mathrm{FL}}$ and assumes values as $C_{\mathrm{green}}^{\mathrm{FL}}$ with height. Accordingly, the parameters $\lambda_{\mathrm{c}}$, $\lambda_{\mathrm{b}}$, and $\lambda_{\mathrm{r}}$ (Fig. 5d), and $R_{605}^{\mathrm{CWL}}$ (Fig. 5e) exhibit BBA values that are almost independent of the state of the atmosphere, but change dramatically at the transition from boundary-layer aerosol to BBA, which highlights the dependence of the spectral shape on aerosol type, or source. The center wavelength of the BBA spectrum amounts to $505\ \mathrm{nm}$ (layer mean), which from our experience is a comparatively small value.

Figure 6 presents the aerosol fluorescence spectra at the heights color-coded in Fig. 5. Focus is on BBA, the spectrum of the boundary-layer aerosol at $2.1\ \mathrm{km}$ is depicted for comparison. From top to bottom, spectra of spectral fluorescence backscatter coefficient ($\beta^{\mathrm{FL}}$), spectral fluorescence capacity ($\beta^{\mathrm{FL}}/\beta_{\mathrm{par}}$), and spectral fluorescence capacity ratio are shown. The latter form of representation was chosen in order to better visualize the height dependence of the aerosol fluorescence spectrum. The reference height can be freely selected, in this case it is $3.6\ \mathrm{km}$, which is the height of the filament at the lower boundary of the BBA layer.

Spectra of $\beta^{\mathrm{FL}}$ (Fig. 6a) and capacity (Fig. 6b) exhibit a similar shape, but the capacity spectra are closer together, implying that some of the variability seen in Fig. 6a is due to changes in the number density of the fluorophores rather than changes in their inelastic properties. The BBA spectrum is broad and has no distinctive features. In this measurement, it peaks between $505$ and $518\ \mathrm{nm}$. From its maximum it slopes down to shorter and longer wavelengths reaching similar values at the far ends of the spectrometer measurement range. The drop on the blue shoulder is faster than on the red shoulder, but overall the BBA spectrum can be described as quite symmetrical. With height, the spectrum becomes increasingly Gaussian without, in this case, the central wavelength shifting. In contrast, the spectrum of boundary-layer aerosol is asymmetrical and can therefore be clearly distinguished from the BBA spectrum. The maximum lies at blue colors (here at $441\ \mathrm{nm}$), wavelengths so short that $\lambda_{\mathrm{b}}$ cannot be determined because it falls outside the spectrometer range (Fig. 5d). The long-wavelength shoulder declines slowly and nearly evenly.

Interpretation of the spectra of spectral fluorescence capacity ratio is quite intuitive (Fig. 6c). The deviation of a ratio curve from the reference line (horizontal line at the value of 1) indicates how much the capacity spectrum under consideration differs from the reference capacity spectrum at a given wavelength. If its value is lower, the spectrum has a lower fluorescence capacity than the reference spectrum, and vice versa. A straight horizontal capacity-ratio curve therefore signifies that the spectral shape of a fluorescence spectrum has not changed compared to the reference spectrum (but fluorescence capacity may have changed).





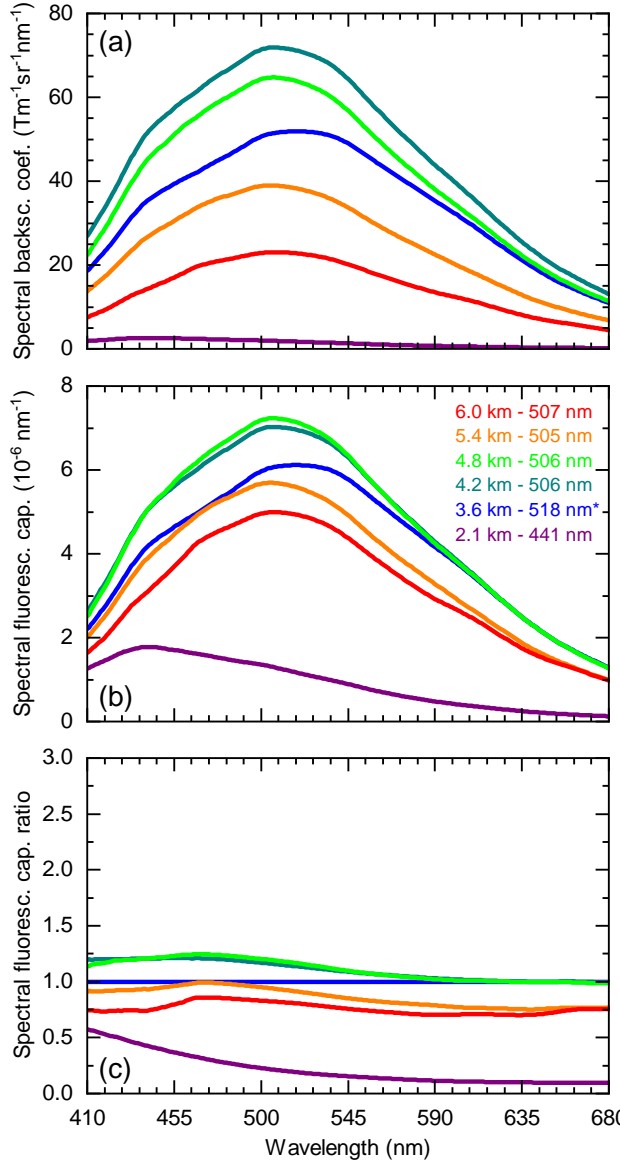

**Figure 6.** Spectral measurements of the BBA layer at 22:30 UTC on 4 July 2023. Different visualizations are chosen to highlight spectrum variability: **(a)** spectral fluorescence backscatter coefficient, **(b)** spectral fluorescence capacity, and **(c)** spectral fluorescence capacity relative to reference spectral fluorescence capacity. Spectra are presented for the heights selected in Fig. 5 (altitude and wavelength of the maximum of the fluorescence spectrum are indicated in panel **b**, reference height is marked with an asterisk).

If the ratio curve rises with wavelength, however, the spectrum has shifted to longer wavelengths (red shift), and if it declines, it has shifted to shorter wavelengths (blue shift).



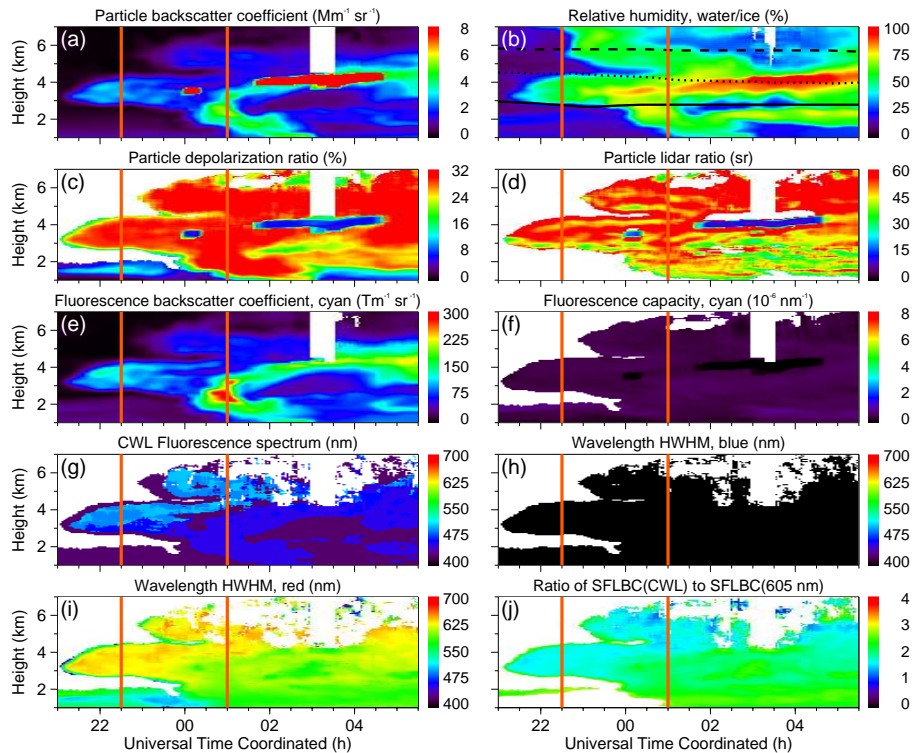

**Figure 7.** Similar to Fig. 4, but for the RAMSES measurement in the night of 22–23 February 2021 between 21:00 and 05:30 UTC. Black lines in panel **b** indicate 0 °C, -10 °C and -20 °C isotherms. The measurements at 22:30 and 01:00 UTC (times marked by vertical orange lines) are analyzed in Figs. 8 and 9.

In the case of our measurement example, BBA spectrum variability is very low (within $\pm 25$ % for all wavelengths), which
is quite unusual. Except for a second homogeneous BBA case a few days later on 9–10 July 2023 (Fig. A8), BBA spectral properties generally changed more dynamically with height and time during the other measurement nights in our data set (see Sect. 3.2), or exhibited distinctly layered properties (references to overview plots of measurement examples are given in Tab. 1). In comparison, the fluorescence spectrum of the boundary-layer aerosol is strongly blue-shifted, and fluorescence capacity is low, particularly at long wavelengths.

**3.1.2 Mineral dust**

After BBA, Saharan dust is the most frequent aerosol type that generated significant aerosol events in the free troposphere over Lindenberg, especially in February 2021 and March 2022. The measurement night of 22–23 February 2021, which is displayed in Fig. 7, has been selected from more than 10 cases as a measurement example. At a first glance, the reader might be misled to conclude that the atmospheric conditions were dynamic and that vertical transport of boundary-layer aerosols beginning
at 01:00 UTC ultimately caused the formation of the altocumulus cloud at 4 km at about 02:00 UTC. However, a closer





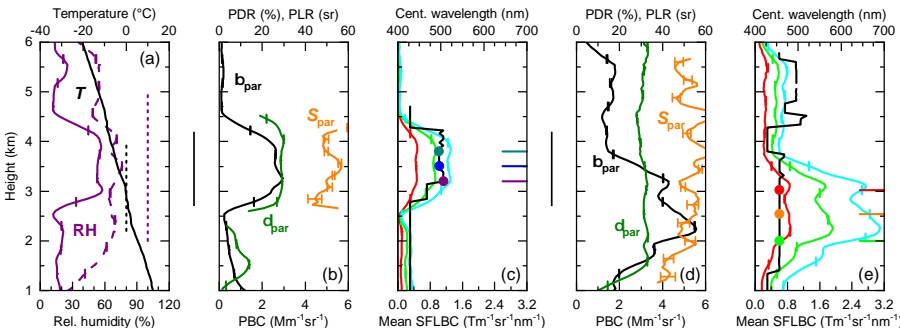

**Figure 8.** Profile measurements of the dust layer at 22:30 and 01:00 UTC in the night of 22–23 February 2021. **(a)** Relative humidity observed at 22:30 (solid violet curve) and 01:00 UTC (dashed violet curve) and temperature (22:30 UTC, the later profile is nearly identical). **(b, d)** Particle backscatter coefficient, particle depolarization ratio and particle lidar ratio, and **(c, e)** mean spectral fluorescence backscatter coefficients for the four spectral bands (colored curves) and wavelength of the maximum of the fluorescence spectrum as measured at 22:30 UTC (panels **b** and **c**) and 01:00 UTC (panels **d** and **e**). Colored symbols and horizontal lines indicate altitudes for which the fluorescence spectra are presented in Fig. 9.

look at the history of the air parcels reveals a different scenario. For this purpose, not only the back-trajectories but also the measurements of the Lindenberg wind radars and the DWD ceilometer network (https://www.dwd.de/EN/research/projects/ceilomap/ceilomap_node.html) were analyzed, which allowed the development of the dust cloud to be traced. According to our analysis, horizontal advection dominated at all times, and vertical transport was absent. Clouds formed along the trajectory,

including the altocumulus we observed (lidar ratios directly above the cloud were rejected by the automated quality control because multiple scattering was not taken into account), occasionally precipitating. Aerosol was washed out as a consequence as maybe best evidenced by the reduced particle and fluorescence backscatter coefficients (blue patches in Figs. 7a and 7e) between approximately 2.2 and 3.4 km starting at 01:15 UTC. Furthermore, it turned out that the dust transported to Lindenberg originated from different desert regions, namely western and central North Africa. Remarkably, the two types of dust can

be easily distinguished by their fluorescence properties; this observation is probably a scientific first. While Saharan dust from mainly Mauritania, Mali and western Algeria predominated in the aerosol plume at the beginning, discernible by longer $\lambda_c$ and $\lambda_r$ (Figs. 7g and 7i) and smaller $R_{605}^{CWL}$ (Fig. 7j), it was displaced upwards over time by dust from mostly central Algeria and Libya (fluorescence spectrum shifted to shorter wavelengths, $R_{605}^{CWL}$ increased).

Figure 8 compares the RAMSES measurements at 22:30 and 01:00 UTC to examine this finding quantitatively in more

detail. For both, temperature and relative humidity are displayed in Fig. 8a; dust elastic and fluorescence properties observed at the earlier measurement time are presented in Figs. 8b and 8c, the results for the later measurement in Figs. 8d and 8e.

Relative humidity increases over time, but has no effect on the aerosol properties. Mean spectral fluorescence backscatter coefficients and particle backscatter coefficient follow similar profiles, so one can expect the spectral fluorescence capacity spectra to be flat and height-independent. While both dust types exhibit practically the same particle depolarization and lidar

ratios ($\sim 30\%$ and $\sim 50$ sr, respectively), they possess different spectral fluorescence properties. At 22:30 UTC, the main





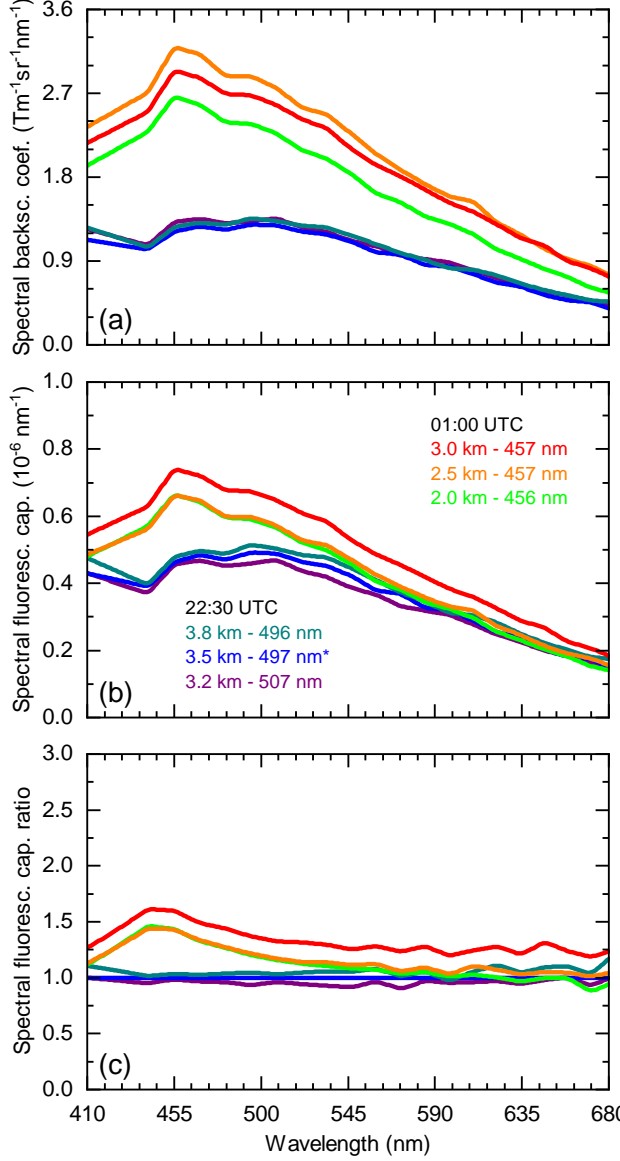

**Figure 9.** Similar to Fig. 6, but for the spectral measurements of the dust layer at 22:30 and 01:00 UTC in the night of 22–23 February 2021.

layer composed of western Sahara dust is characterized by a center wavelength of the fluorescence spectrum around 500 nm. The layer is delimited at the bottom by a filament containing mostly Algerian-Libyan dust, which at approximately 467 nm has a significantly shorter $\lambda_c$. Until 01:00 UTC, the dust plume has broadened significantly. The filament has expanded so that now dust from central North Africa is the dominant type of aerosol below 3.8 km. Its center wavelength has decreased slightly to 457 nm. Western Sahara dust is still observed above 4.3 km, $\lambda_c$ is unchanged.





Figure 9 presents the dust fluorescence spectra for both measurement times. Maximal spectral fluorescence capacity is $< 0.8 \times 10^{-6}$ nm$^{-1}$ in all cases, which is even lower than the boundary-layer value measured on 4–5 July 2023 (Fig. 6b). Capacity spectra of western and central North African dust differ at shorter wavelengths. The former exhibits a slightly rounded, the latter an asymmetric shape, which is similar to what was observed for boundary-layer aerosol, but the spectral maximum

is at longer wavelengths (457 nm) and the red shoulder follows almost a straight line. The differences in dust fluorescence spectra are clearly visible in the ratios of fluorescence capacity when the reference spectrum is taken from the center height of the leading edge of the plume (Fig. 9c). For each dust type, however, spectra and capacities are almost identical. A dependence on the atmospheric state is therefore not apparent. Finally, in view of the discussion in Sect. 3.4 it is worth noting that no cloud effect on the dust fluorescence spectrum was observed either.

**3.2    Fluorescence spectrum of BBA: variability**

The measurement examples in the preceding sections illustrated the dependence of the fluorescence spectrum on the type of the aerosol. What both cases had in common was that their spectra otherwise showed no variability that could indicate other influencing factors such as the state of the atmosphere, the source region or the measurement altitude. In the case of BBA, however, this is the exception rather than the rule, as will be discussed below.

Figure 10 presents the RAMSES measurement of BBA during the night of 26–27 May 2023. The aerosol originated from Western Canada and the transport time was between 95–140 h. In contrast to the measurement on 4–5 July 2023 (Sect. 3.1.1), the BBA layer extends over the entire free troposphere from approximately 3 to 12 km and is not spectrally homogeneous but fragments into numerous filaments with specific spectral properties. In general, a red shift of the spectra with height can be identified, but there are also marked blue shifts, which are layer-dependent (e.g., around 5.3 km between 20:30 and

23:30 UTC). Both features are often to be found in our data set and can be almost called characteristic for spectral BBA measurements (for further measurement examples see Tab. 1 and the overview figures in the appendix). Figure 10 also displays other interesting details. For instance, spectral fluorescence capacity is layer-specific as well, the high values in the filament near the tropopause at 11 km are particularly striking. Since $C_{\mathrm{cyan}}^{\mathrm{FL}}$ is a parameter combining elastic and fluorescence properties of an aerosol it is not possible to tell whether changes in the properties or number concentration of the fluorophore or in those

of the carrier material cause the differences. In the case of the high filament, it is probably both because the BBA spectrum is considerably shifted to longer wavelengths and the particle depolarization ratio is significantly larger than in the lower free troposphere. In fact, with $\delta_{\mathrm{par}} > 15$ % and $S_{\mathrm{par}}$ between 40 and 50 sr it exhibits particle elastic properties that are similar to those measured with RAMSES in the stratospheric BBA layer in August 2017 (not shown). Finally, the increased humidity between 6 and 8 km has no apparent effect on the fluorescence spectra. In general, it has proven difficult to correlate the

fluorescence properties of BBA closely with other variables such as origin, temperature and humidity and particle elastic properties. Related studies can be found in Sect. 3.3.

Figure 11 shows the profiles measured at 21:00 UTC. Below 6 km the atmosphere is dry, the moist layer already mentioned is located above. The particle backscatter coefficient reveals a total of six BBA layers between 2.5 and 11 km at this particular time. The two lowest filaments have only a small vertical extent of approximately 500 m and are clearly separated. BBA layers



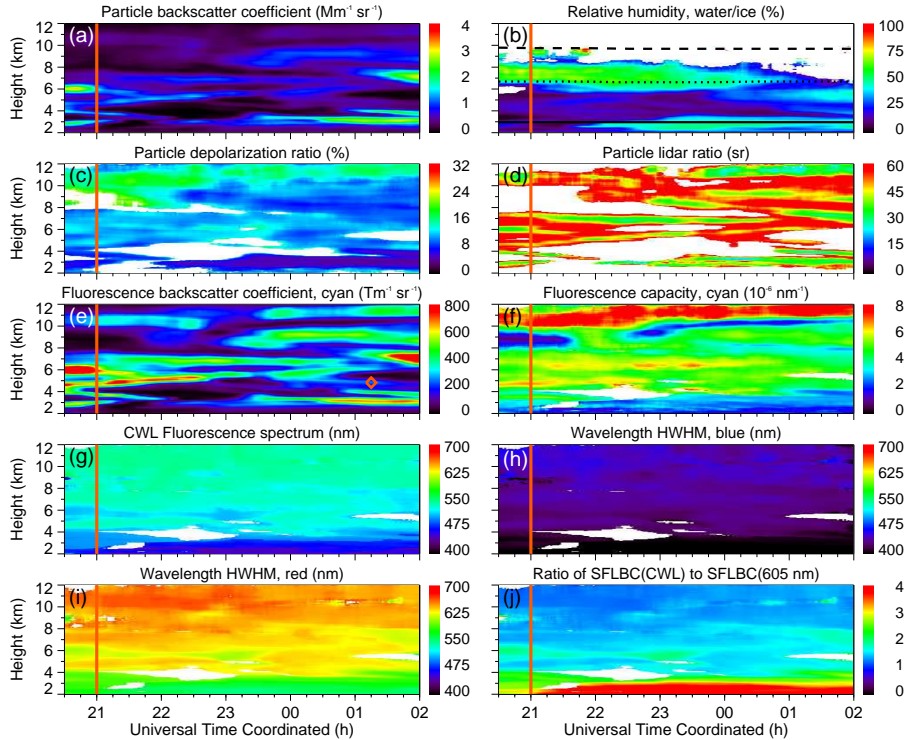

**Figure 10.** Similar to Fig. 4, but for the RAMSES measurement in the night of 26–27 May 2023 between 20:30 and 02:00 UTC. Black lines in panel **b** indicate 0 °C, -25 °C and -50 °C isotherms. The measurement at 21:00 UTC (time marked by vertical orange line) is analyzed in Figs. 11 and 12, the spectrum measured at 4.8 km around 01:15 UTC (orange symbol, panel **e**) is shown in Fig. 1.

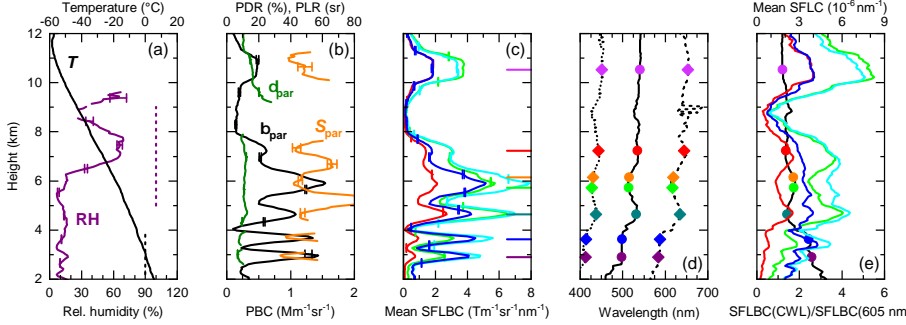

**Figure 11.** Similar to Fig. 5, but for the profile measurement of the BBA layers at 21:00 UTC on 26 May 2023. Colored symbols and horizontal lines indicate altitudes for which the fluorescence spectra are presented in Fig. 12.

3 to 5 lie between 4 and 8 km and adjoin or overlap each other. Layer 6 is located at the tropopause. Although the mean spectral fluorescence backscatter coefficients are correlated with $\beta_{\mathrm{par}}$, they show a different trend with height, resulting in a general increase in spectral fluorescence capacity at longer wavelengths, particularly obvious for the cyan and green false colors.



The relative magnitude of the false color characteristics is also interesting. $\beta_{\mathrm{blue}}^{\mathrm{FL}}$ and $C_{\mathrm{blue}}^{\mathrm{FL}}$ are prominent below 4 km, but decrease with height compared to those of the other false colors and are at the level of, respectively, $\beta_{\mathrm{red}}^{\mathrm{FL}}$ and $C_{\mathrm{red}}^{\mathrm{FL}}$ in the

uppermost filament. The properties evolve most strongly in the cyan and green false color range, the latter dominating near the tropopause. The explanation lies in the general red shift of the BBA fluorescence spectrum with height, which can be clearly seen in the profiles of $\lambda_{\mathrm{c}}$, $\lambda_{\mathrm{b}}$, and $\lambda_{\mathrm{r}}$. An exception, however, is the altitude range around 6 km, where the trend is in the opposite direction, namely towards a shift to shorter wavelengths in BBA layer 4. We have frequently observed such blue shifts, albeit only in specific layers over limited height intervals. A measurement case in which the general trend with height is

towards shorter rather than longer wavelengths has not yet occurred. This observation may indicate that the general red shift in the free troposphere is more related to the atmospheric state than, for example, to the origin and composition of the BBA.

The fluorescence spectra of the six BBA layers are presented in Fig. 12. Their general development with height is obvious in all representations, but the spectra of fluorescence capacity (Fig. 12b) and fluorescence capacity ratio (Fig. 12c) illustrate particularly well that the spectral dynamics is highest at long wavelengths. Near the tropopause, values of almost 3 are found

for the spectral fluorescence capacity ratio at 680 nm, which is in stark contrast to the measurement on 4 July 2023 (see Fig. 6). The capacity-ratio curves for the layers at 4.6 and 7.2 km exhibit a similar wavelength dependence with respect to the reference layer (5.7 km), which illustrates the blue shift of the latter again and, possibly, that it is embedded in a broader, relatively homogeneous BBA field. In view of the profiles shown in Fig. 11, this conjecture is not implausible.

The focus of our studies is on measurements in the free troposphere. However, when looking at the dynamics of the BBA

spectrum, an excursion into the boundary layer is worthwhile. For most of the year, the fluorescence spectrum of the boundary-layer aerosol is similar to the one measured at 2.1 km on 4 July 2023 (Fig. 6). Twice a year, though, at the beginning and end of the gardening season, BBA events can also occur when gardeners traditionally burn their garden waste. At these times, many households also switch to wood-burning fireplaces for heating. Figure 13 shows the fluorescence spectra of two such episodes.

The case of 10 October 2021 is particularly interesting. The fluorescence spectra are very similar to the previously discussed

BBA cases, but they show a rapid decrease in fluorescence capacity and a massive blue shift of about 40 nm with height. Spectral dynamics is significant across the spectrum, and again maximal at longer wavelengths. These changes accelerate between 500 and 700 m. Analyses of high-resolution wind lidar and ceilometer data reveal that the boundary layer was well mixed during the day, and even at night the mixing extended to 400–500 m. Thus, in the first few hundred meters, RAMSES probably measured mainly fresh aerosols from the burning of garden cuttings and fireplace smoke. Above this, a mixture of

these fire aerosols, possibly slightly aged, and background aerosol may have been observed.

On 20 March 2022, the observation provides different results. Here, the BBA spectra change little and retain a shape and center wavelength similar to those in the higher free troposphere (see, for instance, the BBA spectrum measured at 7.2 km in Fig. 12). The two measurement examples underline that the BBA spectrum of the boundary layer strongly depends on the individual case; a deeper understanding would at least require a larger data set and more information about the actual

measurement conditions.





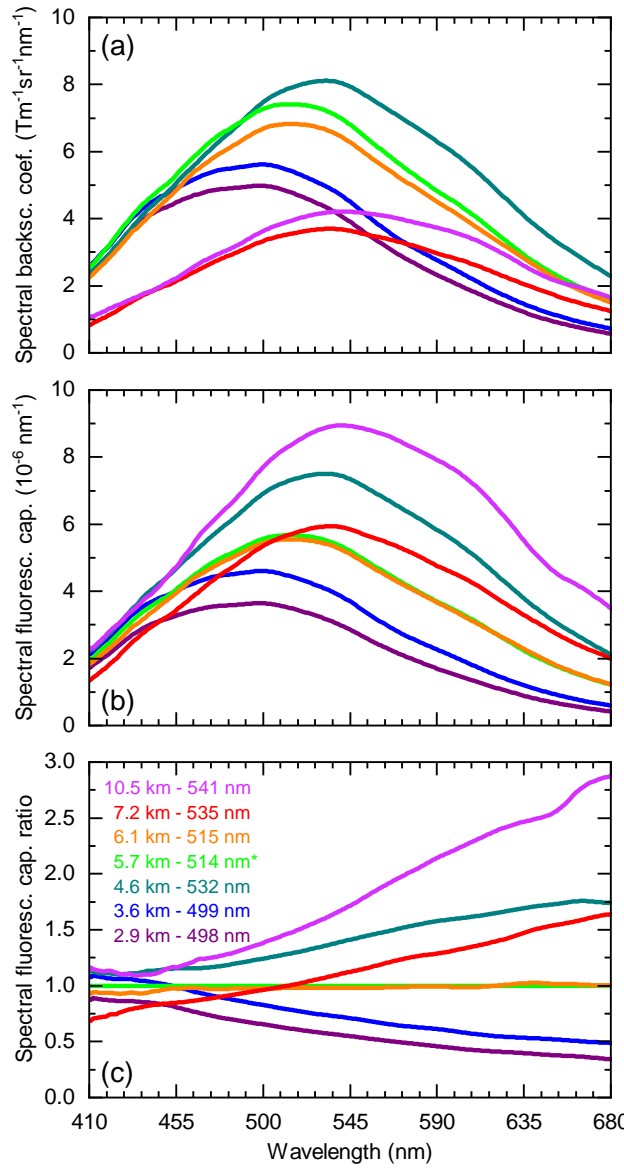

**Figure 12.** Similar to Fig. 6, but for the spectral measurements of the BBA layers at 21:00 UTC on 26 May 2023.

## 3.3 Statistical analyses

So far, the fundamental properties of BBA fluorescence spectra have been described on the basis of individual measurements. However, when it comes to investigating general relationships between the spectral properties and other factors, such as the history of the observed air parcel or the elastic scattering properties of the fluorescing aerosol, this approach is no longer

sufficient and a large data set of high-quality measurements and a complex trajectory analysis are required. In the case of this



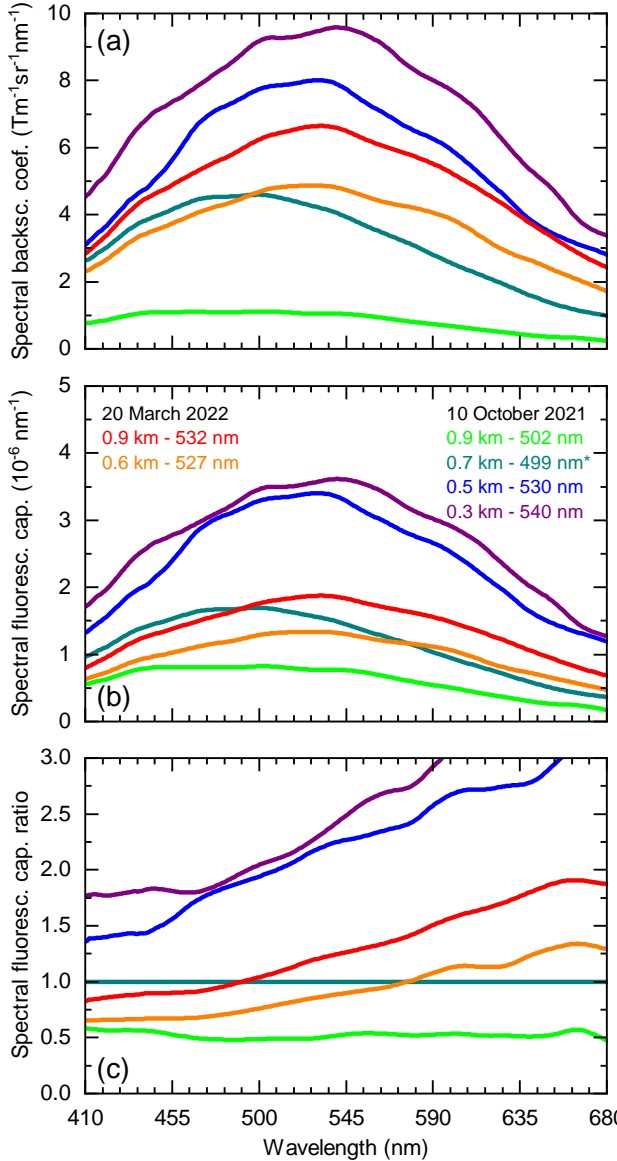

**Figure 13.** Similar to Fig. 6, but for spectral measurements of BBA in the boundary layer at 23:10 UTC on 10 October 2021 and at 19:00 UTC on 20 March 2022.

study, the data set consists of a selection of the best RAMSES measurements from 2020 to 2023 (Tab. 1). The results of our statistical analyses are presented in the following.





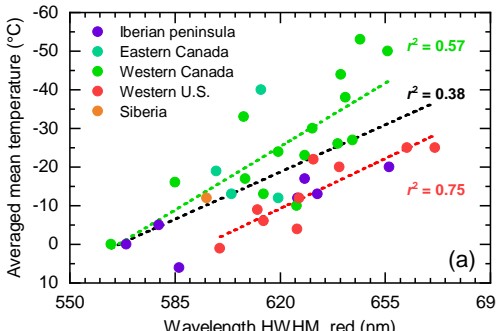
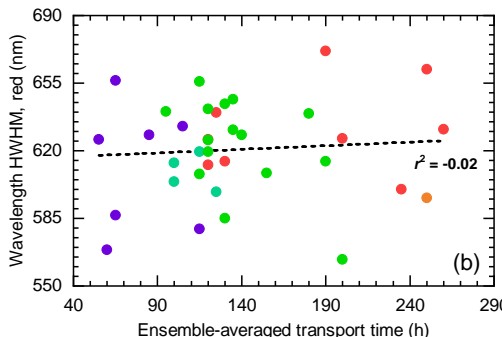

**Figure 14.** Dependence of BBA fluorescence on origin and transport. **(a)** ensemble-averaged mean temperature ($\overline{T}_{\mathrm{mean}}$) as a function of the wavelength of the BBA spectrum half width on the long-wavelength shoulder ($\lambda_{\mathrm{r}}$), and **(b)** $\lambda_{\mathrm{r}}$ as a function of ensemble-averaged transport time ($t_{\mathrm{mean}}$) for the BBA layers listed in Tab. 1. Layer mean values of $\lambda_{\mathrm{r}}$ are used. The data are color-coded according to the major regions of the aerosol sources. The results of linear regression analyses applied to the full data set (black dashed line and $r^2$ value), and (panel **a**) to the Western Canada and Western United States subsets are indicated.

### 3.3.1 Dependence of BBA fluorescence on origin and transport

Figure 14 relates the results of the trajectory analysis summarized in Tab. 1 to the BBA fluorescence spectrum. For its charac-

terization the wavelength of the spectrum half width on the long-wavelength shoulder, $\lambda_{\mathrm{r}}$, was selected as an example, the other spectrum parameters ($\lambda_{\mathrm{c}}$, $\lambda_{\mathrm{b}}$, and $R_{605}^{\mathrm{CWL}}$) exhibit a similar dependence. In Fig. 14a the ensemble-averaged mean temperature ($\overline{T}_{\mathrm{mean}}$) is shown as a function of $\lambda_{\mathrm{r}}$, in Fig. 14b $\lambda_{\mathrm{r}}$ as a function of the ensemble-averaged transport time ($t_{\mathrm{mean}}$), which can be interpreted as the evolution of the fluorescence spectrum with height and time, respectively. The data points are colored according to the major region from which the aerosol particles originated.

Looking at the full data set, it can be seen that the BBA fluorescence has an albeit weak correlation with $\overline{T}_{\mathrm{mean}}$ (Fig. 14a), while it is uncorrelated with $t_{\mathrm{mean}}$ (Fig. 14b). The latter is also true for the mean relative humidity along the trajectory (not shown), but this may also be due to the limited knowledge of the existing humidity field.

However, in the case of the relationship between $\overline{T}_{\mathrm{mean}}$ and $\lambda_{\mathrm{r}}$, a visual inspection suggests that the results could be improved if the data set were split into subsets according to the major wildfire regions. Indeed, if only data points attributed to the Western

United States are considered, the $r^2$ value nearly doubles to 0.75. The same $r^2$ value is found for the Europe subset and for the combination of these two major regions, and also the linear regression lines agree well (not shown). Similarly, the Western Canada subset provides better correlation ($r^2 = 0.57$) than the full data set. Adding the data points of Eastern Canada degrades the correlation coefficient somewhat ($r^2 = 0.50$), but the regression line remains virtually unchanged (not shown).

These findings justify a division of the entire data set into subsets depending on the source region. Since the results for the

Iberian Peninsula and the Western United States on the one hand and for Eastern and Western Canada (and Siberia) on the other are similar and the transport time has only a minor effect, the vegetation or climate zones in which the forest fires rage seem to



determine the BBA spectrum to a certain extent. However, given the uncertainties in the analysis of the back-trajectories and the small-scale variability of the vegetation cover, one cannot expect the correlation to be pronounced.

A comparison of the RAMSES measurements on 1–2 June and 4–5 June 2023 provides a good example in this respect. The back-trajectory analysis indicates that the fire centers were quite close to each other in Western Canada, namely in northern Alberta and in Saskatchewan (Tab. 1), and yet the fluorescence spectra display considerable differences. On 1–2 June 2023 (Fig. A6), the BBA spectrum is at longer wavelengths than on 4–5 June 2023 (Fig. A7) and thus resembles more those from the Western United States. The predominant factor in this case appears to be vegetation rather than geographic location, as in northern Alberta mixed forests prevail while in Saskatchewan mainly coniferous forests flourish.

Finally, it is noteworthy that the regression lines of the data subsets of Western Canada and the Western United States have similar slopes, but are shifted by 30–40 nm from each other (Fig. 14a). While the offset is related to the source region as discussed above, the gradient is an indication of the regularly observed red shift of the BBA spectrum with height. This phenomenon is analyzed in more detail in the following Sect. 3.3.2.

### 3.3.2 Dependence of BBA fluorescence on atmospheric variables

For this analysis, measurement cases had to be selected that covered a wide range of values in both temperature and relative humidity. So, for example, the measurement of 20–21 July 2022 could not be considered despite its excellent data quality and the interesting BBA layering with blue shift, because the relative humidity only varied between 15 and 55 % (Fig. A5). Furthermore, all major aerosol source regions should be taken into account if possible. Eventually, the RAMSES measurements of 11–12 September 2020 (Western United States, Fig. A1), 26–27 May 2023 (Western Canada, Fig. 10), 4–5 July 2023 (mainly Eastern Canada with contributions from Western Canada, Fig. 4), and 9–10 July 2023 (Eastern Canada, Fig. A8) were chosen.

Figure 15 illustrates the characteristic parameters of the BBA fluorescence spectra for these measurement nights as a function of the ambient temperature and humidity. Several observations can be made. Firstly, the contour plots are very different for the individual measurements, which means that there is no universal dependence of the fluorescence spectrum on the atmospheric variables. One reason for this is that several aerosol types, i.e. not only BBA, feature in the image. For instance, at warm temperatures and high relative humidity values, aerosols from the upper boundary layer are very likely to dominate, as can perhaps best be seen from the high ratios of the spectral fluorescence backscatter coefficients at the maximum of the fluorescence spectrum and at 605 nm (Figs. 15n–p).

Secondly, if only the BBA is considered, the relative humidity tends to be higher in the upper than in the lower BBA layers. Thirdly, the differences in the fluorescence spectra of wildfire aerosols originating from different major regions, that were already highlighted in the previous Sect. 3.3.1, are also clearly evident here. The spectral red shift with altitude only occurs in BBA from the west coast of North America, and even here there are significant differences depending on the source (Fig. 15, first and second columns), while BBA from Eastern Canada exhibits essentially height-independent fluorescence spectra (Fig. 15, third and fourth columns). Interestingly, the reddish feature that can be seen in the measurement of 4–5 July 2023 (Figs. 15c, k) can be associated with contributions from west coast BBA. Incidentally, our RAMSES measurements and trajectory analyses revealed that the advection of BBA from Eastern Canada is rather the exception and actually only prevailed in June




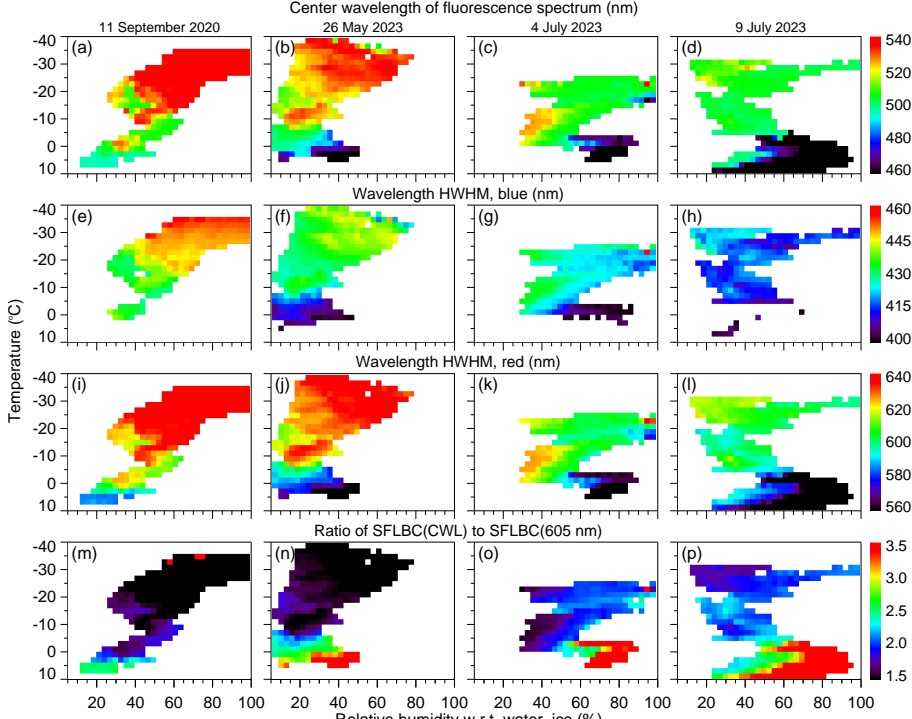

**Figure 15.** Dependence of BBA spectral properties on the state of the atmosphere. **(a–d)** wavelength of the maximum of the fluorescence spectrum (CWL), **(e–h)** wavelength of the spectrum half width (HWHM) on the blue shoulder, **(i–l)** wavelength of the spectrum half width on the red shoulder, and **(m–p)** ratio of CWL to 605-nm spectral fluorescence backscatter coefficients (SFLBC) for the measurement nights of 11–12 September 2020 (panels **a**, **e**, **i**, and **m**), 26–27 May 2023 (panels **b**, **f**, **j**, and **n**), 4–5 July 2023 (panels **c**, **g**, **k**, and **o**), and 9–10 July 2023 (panels **d**, **h**, **l**, and **p**). Data points have bin widths of 3 % relative humidity (absolute) and 2 K, mean values are shown. All measurements from each night were used for the analysis. Minimum and maximum profile heights were determined by the availability of the lidar ratio and the fluorescence-corrected relative humidity, respectively. These requirements resulted in data ranges of 3–9 km between 22:30 and 04:15 UTC, 2.1–8.5 km between 20:15 and 02:15 UTC, 2.1–6.8 km between 20:00 and 02:15 UTC, and 2.1–8.5 km between 20:30 and 01:00 UTC for the four nights. Additionally, only data points with backscatter ratios < 3 were taken into account in order to exclude clouds.

and July 2023 (the transition in the airflow pattern probably occurred between 1 June 2023 and 4 June 2023, see Figs. A6 and A7).

Fourthly, in contrast to the back-trajectory studies in Sect. 3.3.1, Fig. 15 indicates a humidity effect on the BBA fluorescence spectra in the measurements of September 2020 and May 2023: For a given ambient temperature, the wavelength of the

maximum of the fluorescence spectrum (Figs. 15a, b) as well as the wavelengths of the spectrum half width on the blue shoulder (Figs. 15e, f) and on the red shoulder (Figs. 15i, j) increase with relative humidity.



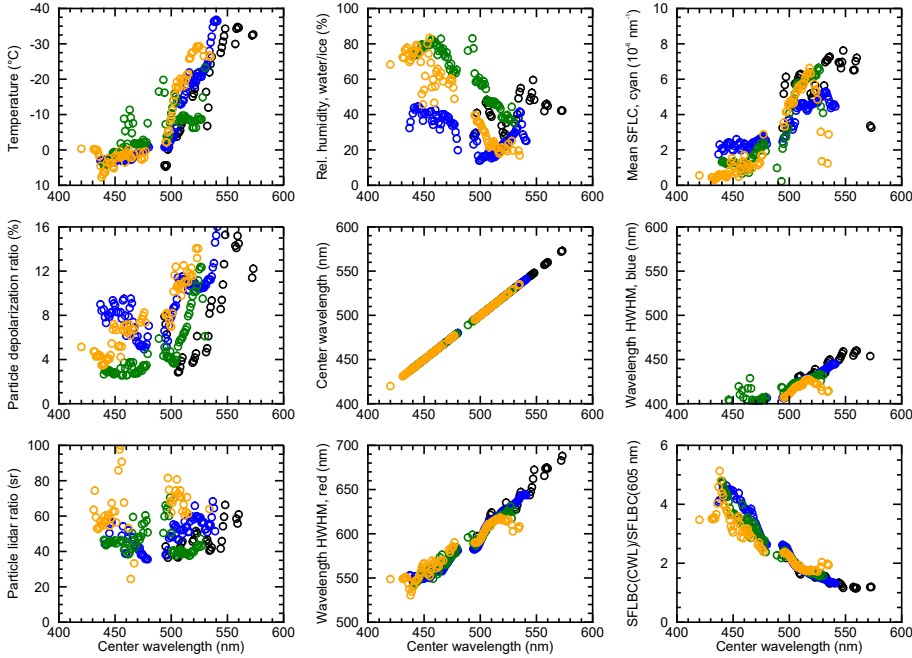

**Figure 16.** Dependence of selected atmospheric variables and elastic and inelastic aerosol properties on the wavelength of the maximum of the fluorescence spectrum. Data points have a bin width of 1 nm, mean values are shown. Measurement nights and data ranges are the same as in Fig. 15, results for 11–12 September 2020, 26–27 May 2023, 4–5 July 2023, and 9–10 July 2023 are illustrated with black, blue, green, and orange symbols, respectively.

Finally, it should be pointed out that the visualization of the BBA spectral properties versus the atmospheric variables also exposes the blue shift anomalies. On 11–12 September 2020, the pattern of a blue shift formed at about 5 km height ($-5\,^{\circ}$C) at 00:45 UTC, with relative humidity reaching values of up to 55 % (Figs. A1g, b). It is discernible as shorter-wavelength

data point clusters, particularly well in Figs. 15a and 15i. The preceding episode with more bluish BBA spectra $< 6.5$–7 km altitude ($-15\,^{\circ}$C) is visible as well. Likewise, the blue shift on 26–27 May 2023 discussed in Sect. 3.2 appears as green data point clusters in Figs. 15b and 15j at temperatures around $-19\,^{\circ}$C and low humidity.

A blue shift with a fundamentally different cause can be observed on 4–5 July 2023 (Fig. 15, third column). It occurs at temperatures around $-17\,^{\circ}$C and high values of relative humidity and probably reflects the effect of an altocumulus on the

BBA fluorescence. This intriguing measurement case is analyzed in Sect. 3.4.2 in detail.

### 3.3.3    Relations between BBA fluorescence, particle elastic properties, and atmospheric variables

In this section, the interrelation between five spectral fluorescence properties of BBA (in addition to $\lambda_{\mathrm{c}}$, $\lambda_{\mathrm{b}}$, $\lambda_{\mathrm{r}}$, and $R_{605}^{\mathrm{CWL}}$ now also $C_{\mathrm{cyan}}^{\mathrm{FL}}$, the mean spectral fluorescence capacity for wavelengths between 455 and 535 nm), its elastic scattering properties ($\delta_{\mathrm{par}}$, $S_{\mathrm{par}}$) and the two atmospheric variables ambient temperature and relative humidity ($T$, RH) are studied for the four





measurement nights. In contrast to the investigations in Sect. 3.3.2, however, only one parameter was considered independent in each case. The statistical analysis was carried out for all nine parameters, but only those for the three with the most significant results are presented. These are the analyses with $\lambda_c$ as an independent variable (representative of the fluorescence properties), $\delta_{par}$, and $T$.

Figure 16 illustrates the dependencies of the aforementioned parameters on the wavelength of the maximum of the BBA

fluorescence spectrum. It should be noted that all data points shown represent mean values of a sufficiently large number of individual values so that they, and thus the trends that emerge, can be regarded as statistically valid. All parameters exhibit an essentially monotonic pattern over $\lambda_c$ without major changes in the gradients. The most striking feature is the high correlation between all spectral parameters and the center wavelength. This applies to $C_{cyan}^{FL}$ as well, even though it is not a solely spectral quantity but a mixture of elastic and inelastic scattering properties. It is also noteworthy that these relationships are independent

of the measurement night, i.e. there is no identifiable dependence on the BBA source region. The same also largely applies to $T$, which is anti-correlated with $\lambda_c$ (once again demonstrating the general red shift of the BBA fluorescence spectrum with height), and to $\delta_{par}$, albeit more weakly. The lidar ratio, on the other hand, bears no correlation.

Surprisingly, relative humidity correlates better with the center wavelength for the measurement days with BBA sources primarily in Eastern Canada (green and orange data points) than for those with western BBA origins (black and blue data

points). However, given the results shown in Fig. 15, this is a misinterpretation due to neglecting the temperature information and not accounting for the presence of BBA contributions from Western Canada.

Ambient temperature and $\delta_{par}$ therefore appear to be the most promising proxy variables for estimating the center wavelength of BBA when spectral fluorescence measurements are not made. If information from a single, discrete fluorescence detection channel were available (Rao et al., 2018) and the fluorescence signal could be calibrated with sufficient accuracy, the spectral

fluorescence capacity could also be considered for this purpose. For further investigation, the various parameters are presented in Figs. A10 and Fig. A11 as functions of the proxy variables $\delta_{par}$ and $T$, respectively. The spectral fluorescence properties show consistent tendencies with both particle depolarization ratio and ambient temperature, but the trends are not independent of the measurement night. Furthermore, significant changes in the gradients occur at $\delta_{par} < 5\ \%$ or $T > -4\ °C$, accompanied by an increase in relative humidity. This effect is not caused by changes in the BBA fluorescence properties but in the type of aerosol:

with warmer temperatures near the freezing level, the probability that boundary-layer aerosol rather than BBA is measured increases significantly, with the corresponding effects on the fluorescence data. So, probably, the observed correlations are not pronounced enough to allow for an estimation of the spectral fluorescence properties of an aerosol layer based on its elastic properties and the ambient conditions.

Lidar ratio is not correlated with $T$ at all (Fig. A11), and only weakly with $\delta_{par}$ in most of the example cases (Fig. A10).

With values between 40 and 70 sr, it remains within the typical range of aged Canadian and Siberian smoke in the troposphere (Haarig et al., 2018). Only the measurement of 9–10 July 2023 is an exception, here $S_{par}$ increases clearly with $\delta_{par}$ and also reaches significantly higher values. Relative humidity (except for 11–12 September 2020) and temperature, on the other hand, are anti-correlated with $\delta_{par}$, with the latter observation in qualitative agreement with the BBA data reported by Haarig et al. (2018).



### 3.4 Cloud interaction and formation

So far, this paper has discussed the fluorescence properties of aerosols as a function of aerosol type and investigated their relationship with the elastic-optical properties of the aerosol, the source region, the transport, and the state of the atmosphere. What has not yet been considered are the effects that an aerosol-cloud interaction could have on the fluorescence properties of aerosols and, conversely, how fluorescence measurements could be used to investigate these interactions. The fact that fluorescence is a powerful means of detecting the presence of aerosols within clouds has been made known by Reichardt et al. (2018), but the observation of coexistence is not yet proof that the aerosol and cloud particles actually interact, are in contact with and affect each other. Whether the aerosol particles are incorporated in the cloud particles or not cannot be deduced directly from the lidar measurements (as can be done, for instance, by examining samples of cloud particles under the microscope) but must be inferred indirectly. Indications of aerosol-mediated cloud formation may be provided, for example, by the profiles of elastic and inelastic particle properties and relative humidity, and the immersion of aerosol particles in cloud particles may be revealed by a suppression of fluorescence or a change in the fluorescence spectrum caused by matrix effects. In the following, these methods are illustrated on the basis of case studies. The effect of the hygroscopic growth of aerosols on the fluorescence spectrum is also discussed using an example measurement.

#### 3.4.1 BBA and cirrus formation

Figure 17 shows the RAMSES measurement of the upper troposphere and lowermost stratosphere on 13–14 September 2020. The prominent feature is an optically thin cirrus layer between 12 and 14 km which is best visible in the volume depolarization ratio (Fig. 17b). The BBA filament located between 13 and 15 km, however, would have probably gone unnoticed with its particle backscatter coefficient near the detection limit (Fig. 17a) and its volume depolarization ratio most of the observation time masked by the cirrus, if the aerosol fluorescence data had not been available (Fig. 17c). Incidentally, the BBA layer is detached from the main BBA event in the lower and middle troposphere which is displayed in Fig. A2.

Information on the relative humidity is required to investigate a possible aerosol effect on cloud formation. Unfortunately, the fluorescence of the BBA filament impairs the water vapor mixing ratio measurements with RAMSES even when the far-range water vapor Raman channel with its ultra-narrow bandwidth of 0.25 nm is used. As a substitute, the measurement with the operational RS41 radiosonde launched on site at 22:45 UTC is considered. Up to the lower edge of the BBA layer the RAMSES night-averaged mixing ratio profile agrees well with the sonde measurement, thus it can be considered meaningful for the lidar observations.

Figure 18 combines the RAMSES and RS41 profiles at a time the radiosonde traversed the cirrus layer. Below the hygropause at about 14 km, relative humidity increased rapidly and reached saturation or slight supersaturation with respect to ice between 13 and 13.8 km before dropping off again. Growth in particle backscatter coefficient accelerated when relative humidity approached 100 %, and particle depolarization ratio and lidar ratio acquired values typical of cirrus (about 45 % and 10 sr, respectively; note that lidar ratio was not corrected for multiple scattering and so the measured value is only about half the true value). With respect to the cirrus cloud, the BBA layer is shifted upwards but overlaps with its upper part. Interestingly, the





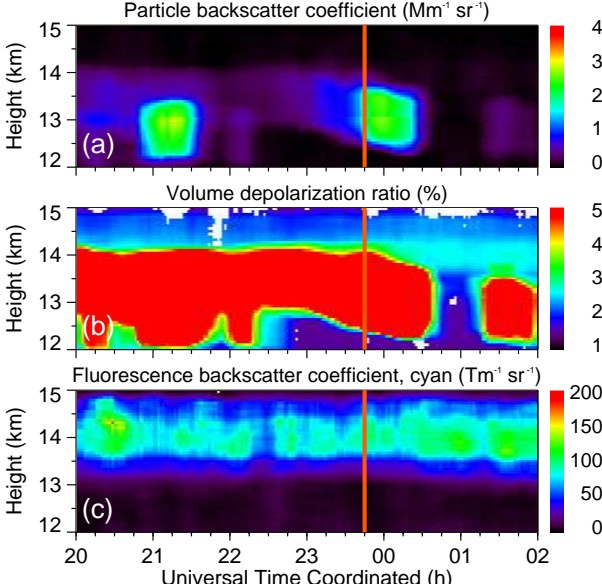

**Figure 17.** Coexistence of a BBA filament and an optically thin cirrus cloud at the tropopause. Temporal evolution of **(a)** particle backscatter coefficient, **(b)** volume depolarization ratio, and **(c)** fluorescence backscatter coefficient (cyan false color: spectrum integrated from 455 to 535 nm) as measured with RAMSES in the night of 13–14 September 2020 between 20:00 and 02:00 UTC. The data shown are a section of the night-long measurement (Fig. A2). The measurement at 23:45 UTC (time marked by vertical orange line) is presented in Fig. 18.

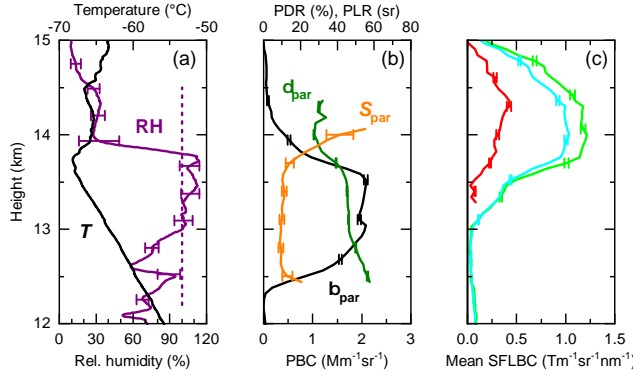

**Figure 18.** Measurement at 23:45 UTC on 13 September 2020. Profiles of **(a)** relative humidity with respect to ice (RH; saturation level indicated by vertical dashed violet line) and temperature ($T$), **(b)** particle backscatter coefficient (PBC, $\beta_{\mathrm{par}}$), particle depolarization ratio (PDR, $\delta_{\mathrm{par}}$) and particle lidar ratio (PLR, $S_{\mathrm{par}}$; not corrected for multiple scattering), and **(c)** mean spectral fluorescence backscatter coefficients (SFLBC) for the cyan, green, and red spectral bands. $T$ and RH profiles were measured with the routine RS41 radiosonde launched at 22:45 UTC on site. Bars indicate statistical measurement errors (RAMSES) or total measurement uncertainty (RS41).



spectral fluorescence backscatter coefficients declined with particle backscatter coefficient, which was particularly significant near the backscatter maximum. This phenomenon could be traced over the entire duration of the measurement, which is why it is probably not random but suggests a relation between the two measurement parameters.

The relative humidity was 113 % maximum with a total measurement uncertainty of 9 % ($2\sigma$), but was mostly so low that the saturation level was within the error limits. This finding suggests that the ice particles readily formed in the presence of BBA particles at very low supersaturation which can be regarded as an indication that nucleation was indeed aerosol-mediated.

The measurement presented in Fig. 17 is just one illustrative example that suggests seeding of clouds by BBA. In fact, systems of high cirrus clouds and partly overlapping BBA layers are quite common in our data archive, and many measurement cases are available for further analysis. The example also demonstrates that fluorescence is an exceptionally sensitive indicator of the presence of BBA particles that would otherwise be undetectable even with powerful multiwavelength lidars. A combined data set consisting of elastic and inelastic particle properties could therefore contribute to the testing of existing nucleation models.

### 3.4.2 Cloud effect on BBA fluorescence spectra

As already indicated and briefly discussed in Sects. 3.1.1 and 3.3.2, the measurement of 4–5 July 2023 could be interpreted as evidence of a cloud effect on the fluorescence spectrum of the BBA aerosols and thus of a direct particle interaction. This hypothesis will be investigated in more detail below. The main difficulties in this analysis are twofold: Firstly, the evaluation of the measurement itself is challenging because the fluorescence backscatter coefficients are small and the signal errors high due to the significant light extinction in the presence of the cloud. Secondly, it needs to be proven that the observed spectral differences were most probably caused by immersion of the BBA particles in the cloud particles and are not the result of transport processes.

Figure 19 displays the RAMSES measurement at 01:36 UTC on 5 July 2023 (cf. Fig. 4). According, to the particle backscatter coefficient the lower cloud edge was at 6.2 km, but elevated values were observed down to 5.7 km. Ambient temperature was around $-20\,^{\circ}$C, and relative humidity was $> 80$ % above 5.4 km and reached levels of slight supersaturation with respect to ice about 300 m below the cloud base. Cloud phase is difficult to retrieve. The UVA spectrometer indicates ice, but this is not certain due to the lower spectral resolution of the spectrum compared to the water spectrometer which is normally used for phase determination but was not operated at the time. The lidar ratio corrected for multiple scattering is around 15 sr which would tend to support the assumption of liquid water, but particle depolarization ratios are too high for a pure water cloud unless significant multiple scattering occurred. The Lindenberg Cloudnet retrieval classifies the layer $< 6$ km as ice (which is certainly not correct based on the particle lidar and depolarization ratios), and the cloud itself as consisting of cloud drops or ice and supercooled droplets. The data are therefore not conclusive, but it seems plausible to assume that a mixed-phase cloud was observed.

Intriguingly, the fluorescence measurement parameters change systematically with height. Unchanged up to approximately 5.5 km, the mean spectral fluorescence backscatter coefficients and capacities begin to decrease in the transition zone up to





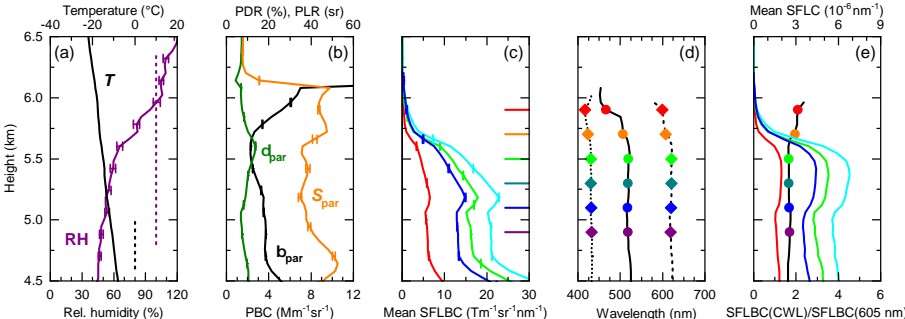

**Figure 19.** Similar to Fig. 5, but for the RAMSES measurement at 01:36 UTC in the night of 4–5 July 2023 (Fig. 4). Colored symbols and horizontal lines indicate center altitudes of height intervals for which averaged fluorescence spectra are presented in Fig. 20.

the cloud, and the spectrum shifts towards shorter wavelengths (blue shift). These trends appear to accelerate in the immediate vicinity of the cloud at high humidity. Within the cloud, the fluorescence fades away almost completely.

To study the effects on the BBA fluorescence spectrum in more detail, it was necessary to average the spectra over 200-m height intervals and 45 minutes in order to reduce the statistical errors. The results are presented in Fig. 20. The sharp decrease
in the spectral fluorescence backscatter coefficients (Fig. 20a) with an approximately unchanged particle backscatter coefficient (Fig. 20b, legend) leads to a drastic decrease in the spectral fluorescence capacity — which could be interpreted as fluorescence quenching due to changes in the molecular environment of the fluorophores. The same applies to the blue shift of the BBA spectrum with height towards the cloud, which can be well seen in the spectra normalized at 508 nm in Fig 20c.

However, this conclusion is not as straightforward as it seems when the atmospheric dynamics of the event are taken into
account. Our trajectory analysis for 01:00 UTC suggests that the trajectories with their end heights at cloud levels had a low probability to pass over wildfires following a slightly different advection pattern than those terminating around 5 km. Thus we cannot exclude the possibility that not cloud processing of BBA spectrally identical to those in the layer below the cloud was observed but some fluorescence of BBA from other sources.

To further strengthen the hypothesis of a cloud effect on the BBA fluorescence spectrum, another RAMSES measurement is
discussed as a second example. The measurement was performed in the night from 30 September to 1 October 2023, Fig. A9 provides a depiction of this complex aerosol and cloud event. Over time, dry air masses carrying BBA from Western Canada made way for humid air with clouds but in this case without any aerosols, as the measurement shows and the trajectory analysis confirms. The altitude ranges in which cloud- and BBA-bearing air masses eventually mixed are indicated by the black areas of the spectral fluorescence capacity (Fig. A9f). It is within these regions where the BBA fluorescence spectrum experiences a
blue shift, as indicated by the discolored fringes of the spectrum-related properties in Figs. A9g–A9j. The effect is particularly pronounced (CWL is shifted by 25–35 nm to shorter wavelengths) in the three BBA filaments stretching towards the cloud after 22:30 UTC, and it is more related to the presence of cloud particles than to water vapor saturation, since the onset of the blue shift correlates better with the lower cloud edge than with the 100 % relative humidity level.





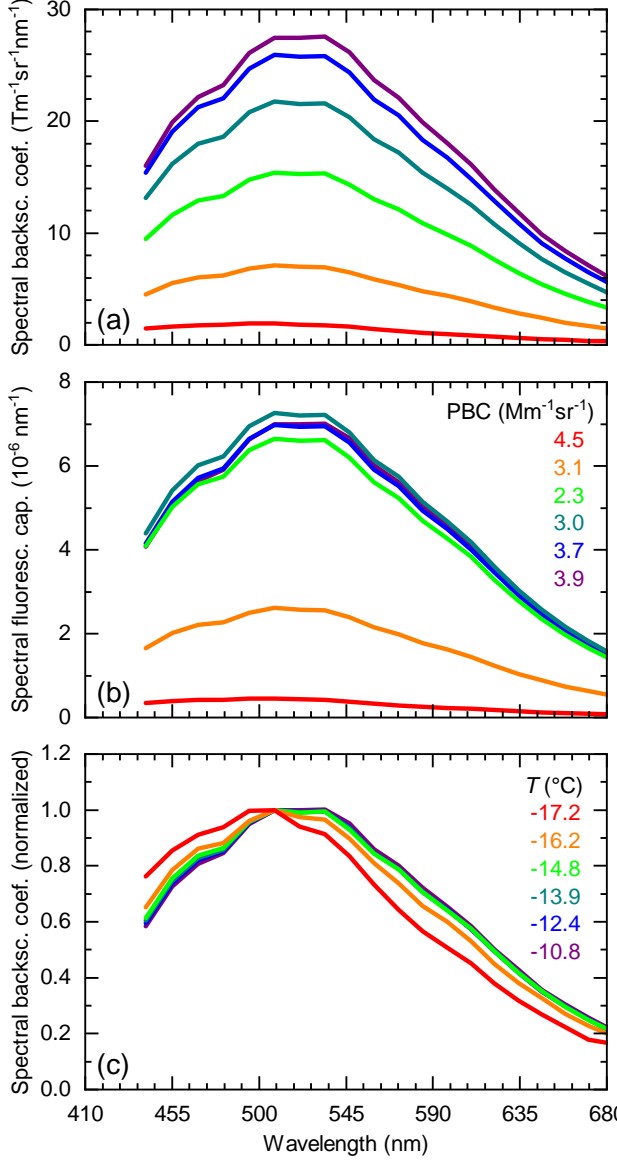

**Figure 20.** Cloud effect on BBA fluorescence spectra measured with RAMSES in the night of 4–5 July 2023. **(a)** Spectral fluorescence backscatter coefficient, **(b)** spectral fluorescence capacity, and **(c)** spectral fluorescence backscatter coefficient normalized at 508 nm. Shown are mean spectra, averaged over 200-m height intervals (center altitudes indicated in Fig. 19) and over the period from 01:30 to 02:15 UTC. Mean particle backscatter coefficient and temperature of the layers are listed in panels **b** and **c**, respectively.

According to the Cloudnet target classification, the cloud system was an ice cloud with an embedded layer of supercooled
water droplets descending over time, which is consistent with the Raman spectra of the cloud water measured with the UVA



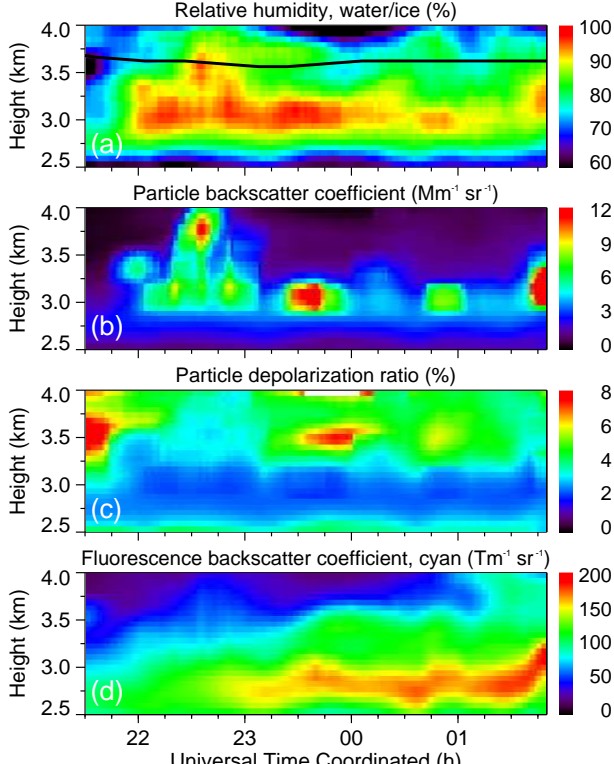

**Figure 21.** Temporal evolution of **(a)** relative humidity (with respect to water and ice above and below 0 °C, respectively; 0 °C isotherm indicated by black line), **(b)** particle backscatter coefficient, **(c)** particle depolarization ratio, and **(d)** fluorescence backscatter coefficient (cyan false color: spectrum integrated from 455 to 535 nm) as measured with RAMSES in the night of 9–10 July 2023 between 21:30 and 01:50 UTC. The data shown are a section of the night-long measurement (Fig. A8).

spectrometer. Remarkably, the blue shift of the BBA fluorescence spectra at all times occurred in the height range with the supercooled droplets, suggesting that these were in fact the sites of the observed cloud processing.

In summary, the measurements discussed here, and especially the one performed at the end of September 2023, present the clearest indication to date of an interaction between clouds and fluorescent aerosols. It is very likely that supercooled water
droplets played a significant role here. It cannot be ruled out, however, that ice particles could also have an effect on the fluorescence of aerosols, but no convincing evidence has yet been found. Although the measurement of 19–20 September 2020 suggests such an ice effect, the observed stratification of the BBA fluorescence could also be of dynamic origin (Fig. A3). Finally, it should be noted that the detection of matrix effects is only a sufficient condition for the existence of an aerosol-cloud interaction, not a necessary one. It is quite conceivable that the inclusion of an aerosol particle in a cloud particle only leads to
changes in the fluorescence spectrum under certain conditions and not always.





### 3.4.3 Effect of water uptake on aerosol fluorescence spectra

In the following, an example is discussed to show that changes in the microphysical properties of aerosols do not necessarily have an effect on their fluorescence spectra. Figure 21 displays the temporal evolution of an aerosol layer in the lower free troposphere as measured with RAMSES on 9–10 July 2023. Within this measurement segment, the relative humidity varies

between approximately 70 and almost 100 %. In general, the particle backscatter coefficient increases with relative humidity while the particle depolarization ratio decreases, indicating that the aerosol particles grew larger as they took up water. In contrast, the aerosol fluorescence does not appear to correlate with the particle changes at first glance.

In order to gain further clues, the fluorescence spectrum was examined as a function of the particle backscatter coefficient. For this purpose, seven intervals spanning the range between 0.5 and 13 $\mathrm{Mm^{-1}sr^{-1}}$ were defined and the fluorescence spectra

within the time-height range shown in Fig. 21 correspondingly clustered and analyzed. The results are presented in Fig. 22, from which several conclusions can be drawn. First, the shape of the spectrum indicates that the fluorescence did not originate from BBA. Rather, it is similar to the spectrum of the boundary-layer aerosol shown in Fig. 6, although the spectral maximum of the latter is at shorter wavelengths and the spectral fluorescence capacity higher. There are also similarities to the fluorescence spectrum of Saharan dust measured at 01:00 UTC on Feb. 23 (Fig. 9), however, spectral fluorescence capacities differ

at longer wavelengths. So probably, mineral-rich boundary-layer aerosol was observed. Secondly, the spectral fluorescence backscatter coefficients are relatively independent of the particle backscatter coefficient (Fig. 22a), which means that the spectral fluorescence capacity decreases approximately in proportion to the particle backscatter coefficients (Fig. 22b). This finding indicates that fluorescence quenching did not occur upon water uptake. Finally, the fluorescence spectra normalized at 534 nm show that there is also no significant effect on the shape of the spectrum (Fig. 22c). For wavelengths > 500 nm the spectra are

identical. There are slight variations at shorter wavelengths, which could either be due to water uptake or a temperature effect, but statistically these differences are irrelevant (the standard deviations are greater than the discrepancies).

In conclusion, in this specific measurement case aerosol particle growth left the fluorescence properties unaffected. Probably, this is to be expected if the fluorescence originated from insoluble minerals as was quite likely the case here. An alternative explanation could be that actually a mixture of different aerosol types was observed, of which one contained soluble substances

and took up the water (similarly to the model of Lewis et al., 2009) while another carrying the fluorophores did not, and so the molecular environment of the fluorophores did not change.

## 4    Summary and perspectives

In this study we have summarized the results of 14 years of fluorescence measurements on aerosols in the free troposphere with the spectrometric fluorescence and Raman lidar RAMSES at the Lindenberg Meteorological Observatory, Germany. The focus

was on results that could only be obtained by spectrometric measurements, i.e. for which the detection of fluorescence with a single discrete receiver channel is not sufficient. It is to be understood as a fundamental paper which, based on observations, shows connections between aerosol fluorescence and many other parameters and influencing factors, such as the elastic-optical particle properties, the type and origin of the aerosols, the atmospheric state variables and clouds, and thus raises questions





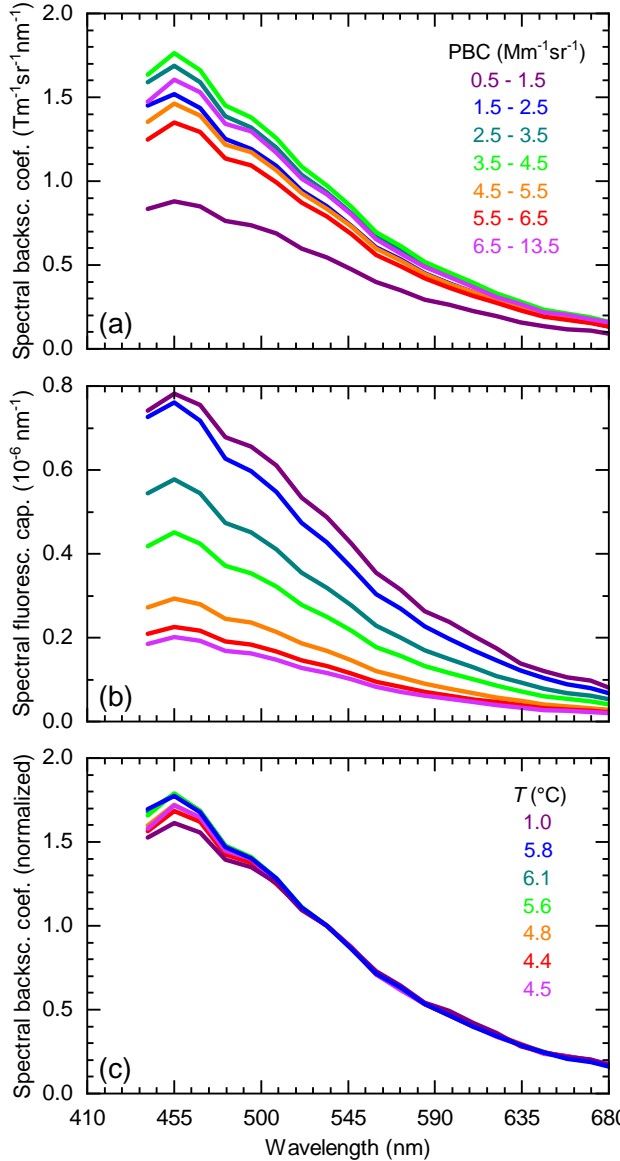

**Figure 22.** Effect of water uptake on aerosol fluorescence spectra measured with RAMSES in the night of 9–10 July 2023. **(a)** Spectral fluorescence backscatter coefficient, **(b)** spectral fluorescence capacity, and **(c)** spectral fluorescence backscatter coefficient normalized at 534 nm. Shown are mean spectra, sorted according to the particle backscatter coefficient intervals indicated in panel **a** and averaged over the time-height segment of the RAMSES measurement displayed in Fig. 21. Mean temperature of the data subsets is listed in panel **c**.

which should inspire further investigations. The following perspectives were identified as relevant for aerosol and atmospheric

research:



(a) How close is the relationship between the fluorescence spectrum and the aerosol source? Our investigations provide evidence for both BBA and Saharan dust that the fluorescence spectra may contain more information about the fluorescing compounds than just their substance class. A refinement of aerosol typing thus seems possible. More measurements, preferably at different sites, combined with meticulous back-trajectory analyses and the use of additional data (as in our study, for example) are needed.

(b) How does cloud processing affect the fluorescence spectrum of aerosols, and for which clouds and aerosols does an effect occur? The blue shift of the BBA fluorescence spectrum in cloud layers with supercooled water droplets discussed in this study is the first lidar-derived evidence that aerosol and cloud particles not only coexist but actually interact.

(c) What effect does BBA have on cloud formation? As shown previously by Reichardt et al. (2018), Gast et al. (2024) and in the present study, fluorescence is an extremely sensitive means of detecting aerosols in the tropopause region and provides information on the nucleation of ice clouds. This question must be addressed, as it is of great importance for cloud modeling.

(d) Which molecules or substances are actually causing the fluorescence? The determination of the fluorophores is well beyond the scope of this study. In this context, laboratory investigations are required (ideally with a parameter set similar to that of Gomez et al., 2018), which are already under way (Li et al., 2019; Richardson et al., 2019; Saito et al., 2022), as well as what could be called chemical lidars (Tatarov and Müller, 2021).

(e) How can the observed red shift of the BBA fluorescence spectra with altitude (or temperature) in layers of the same origin be explained? Although examples of red and blue shifts with temperature are known from the literature (Kundu et al., 2019; Deepa et al., 2020; Schweizer et al., 2021), the chemical compounds investigated and the ambient conditions in these studies cannot be regarded as representative for aerosol measurements in the atmosphere. Experiments in aerosol chambers would be required to answer this question.

(f) Is there a relation between the morphology of the BBA particles or the molecular environment of the fluorophores and the aerosol fluorescence and does water uptake by hygroscopic constituents play a role? Our BBA measurements indicate that particle depolarization ratio of all elastic-optical particle properties correlates best with the spectral fluorescence properties, but with relative humidity this is not the case. Generally, $\delta_{\mathrm{par}}$ of BBA increases with height. Near the tropopause and certainly within the stratosphere it reaches values that require the carrier material of the fluorophores to have a certain size and irregular shape. So smaller $\delta_{\mathrm{par}}$ values at lower altitudes would have to go hand in hand with either a collapse in size or splitting of the particles, or significant water uptake so that the shape becomes spherical. That the latter is not always the case, demonstrates, for instance, the measurement on 19–20 September 2020 (Fig. A3) where near the tropopause relative humidity is high and $\delta_{\mathrm{par}}$ still pronounced. Moreover, the measurement on 26–27 May 2023 reveals that $\delta_{\mathrm{par}}$ does not necessarily respond to a sharp increase in relative humidity (Fig. 11). Furthermore, calculations by Bi et al. (2018) show that wetted irregular particles have even increased particle depolarization ratios, only if fully immersed in a droplet $\delta_{\mathrm{par}}$ is effectively suppressed. In view of these observations, it therefore seems premature

to explain the low $\delta_{\mathrm{par}}$ values of BBA in the lower free troposphere with a spherical shape due to water uptake of the
particles and to associate the morphological change with bluer BBA fluorescence spectra.

Finally, it should be noted that the use of a second, shorter excitation wavelength would offer a significant extension of the experimental capabilities. Such a spectrometric 2-wavelength fluorescence lidar would enable excitation-emission spectroscopy of atmospheric aerosols in its simplest form, which should contribute to improved differentiation between aerosol types. Initial field measurements and accompanying laboratory investigations have already been carried out by Li et al. (2019); Richardson
et al. (2019); Saito et al. (2022), however, the use of the fourth harmonic generation wavelength of the Nd:YAG laser at 266 nm led to a strict limitation in range. These instruments are therefore not suitable for studying aerosols in the free troposphere. Lidars that are operated with excitation wavelengths of 308 nm and between 351–355 nm, such as the standard stratospheric ozone DIALs in the Network for the Detection of Atmospheric Composition Change (NDACC, De Mazière et al., 2018), would be more promising here and well suited for implementation. The subsystems should be operated alternately to prevent
the backscattered atmospheric spectra from overlapping.

*Data availability.* Data presented in this paper are not publicly available at this time but may be obtained from the authors upon reasonable request.

*Author contributions.* JR conceptualized the spectrometric investigation, designed the lidar, developed the methodology, analyzed the data, and wrote the manuscript. FL conducted the back-trajectory analyses and helped with the preparation of the paper. OB was responsible for
all technical aspects of RAMSES and performed the measurements.

*Competing interests.* The authors declare that they have no conflict of interest.

*Acknowledgements.* The authors acknowledge the NOAA Air Resources Laboratory (ARL) for the provision of the HYSPLIT transport and dispersion model and READY website (https://www.ready.noaa.gov). The authors further acknowledge the use of data from NASA's Fire Information for Resource Management System (FIRMS) (https://www.earthdata.nasa.gov/data/tools/firms), part of NASA's Earth Science
Data and Information System (ESDIS). In this publication, standard fire products from the Moderate Resolution Imaging Spectroradiometer (MODIS) were used.



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



**Appendix A:  Supplementary measurements and analyses**

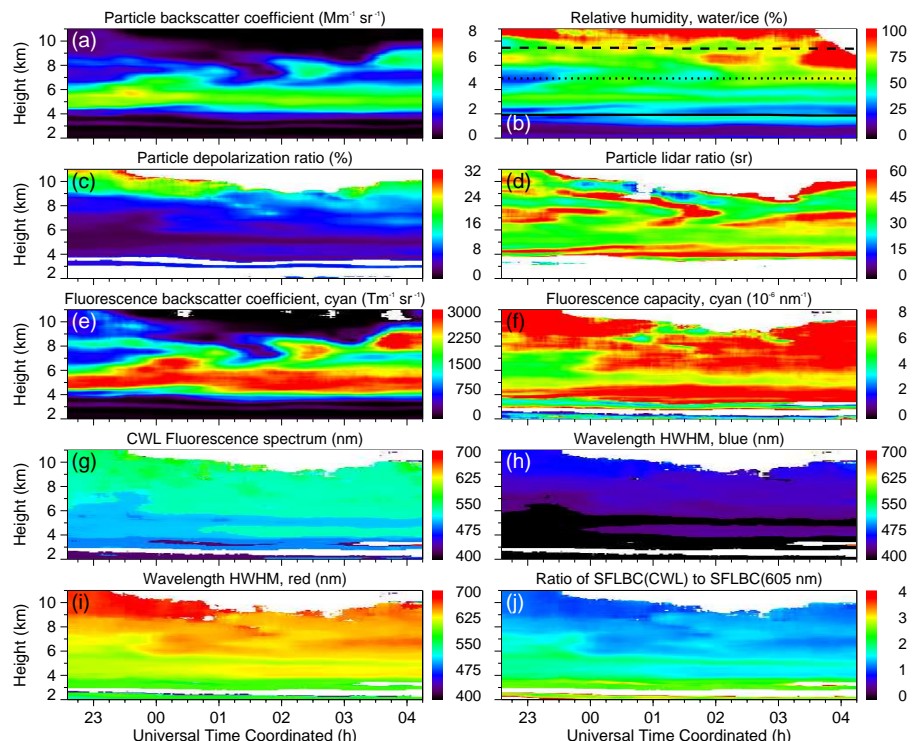

**Figure A1.** RAMSES measurement in the night of 11–12 September 2020 between 22:36 and 04:15 UTC. Temporal evolution of **(a)** particle backscatter coefficient, **(b)** relative humidity (with respect to water and ice above and below 0 °C, respectively; 0 °C, -20 °C and -40 °C isotherms indicated by black lines), **(c)** particle depolarization ratio, **(d)** particle lidar ratio (not corrected for multiple scattering), **(e)** fluorescence backscatter coefficient (cyan false color: spectrum integrated from 455 to 535 nm), **(f)** spectral fluorescence capacity (mean value, 455–535 nm), **(g)** wavelength of the maximum of the fluorescence spectrum (CWL), **(h)** wavelength of the spectrum half width (HWHM) on the blue shoulder, **(i)** wavelength of the spectrum half width on the red shoulder, and **(j)** ratio of CWL to 605-nm spectral fluorescence backscatter coefficients (SFLBC). For each profile, 1200 s of lidar data are integrated, the calculation step width is 120 s. The resolution of the raw data is 60 m, signal profiles are smoothed with a sliding-average length increasing with height. White areas indicate where data were rejected by the automated quality control process.



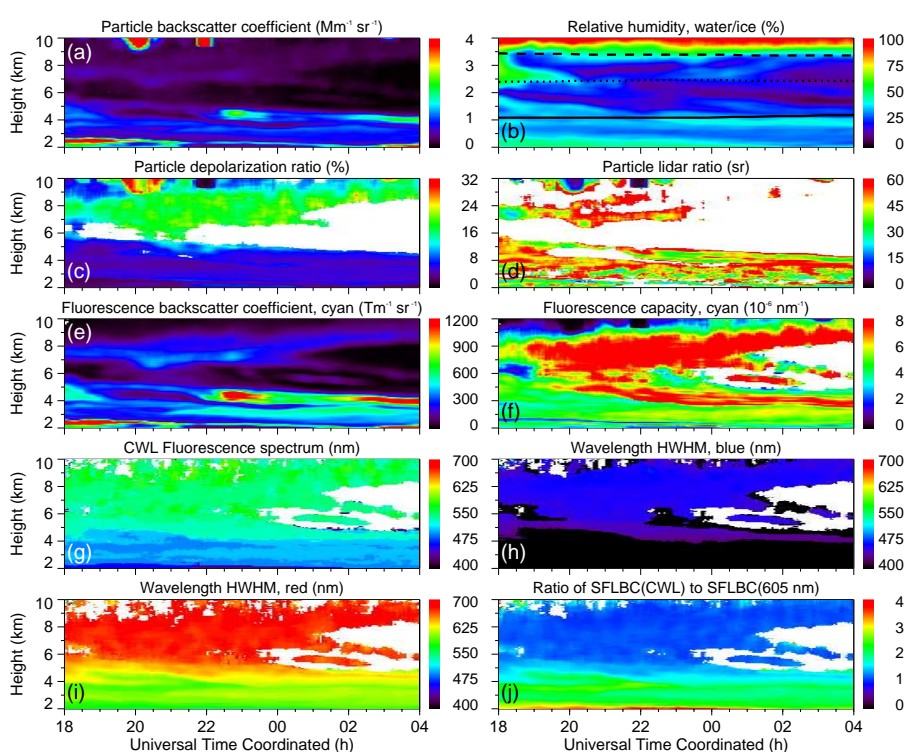

**Figure A2.** Similar to Fig. A1, but for the RAMSES measurement in the night of 13–14 September 2020 between 18:00 and 04:00 UTC. Black lines in panel **b** indicate 0 °C, -15 °C and -30 °C isotherms.



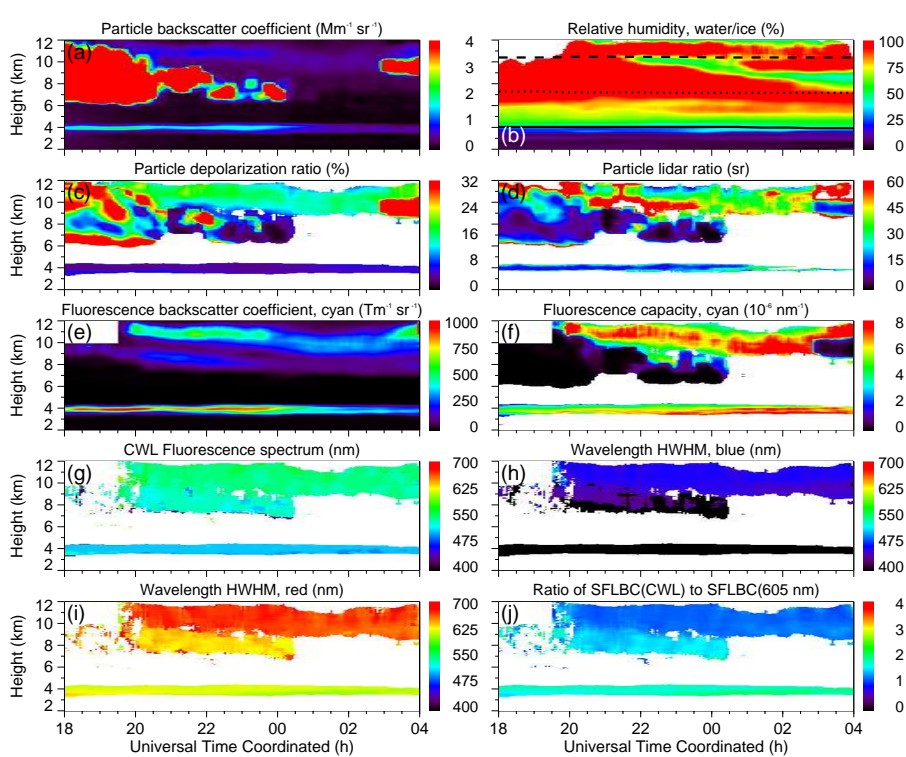

**Figure A3.** Similar to Fig. A1, but for the RAMSES measurement in the night of 19–20 September 2020 between 18:00 and 04:00 UTC. Black lines in panel **b** indicate 0 °C, -25 °C and -50 °C isotherms.



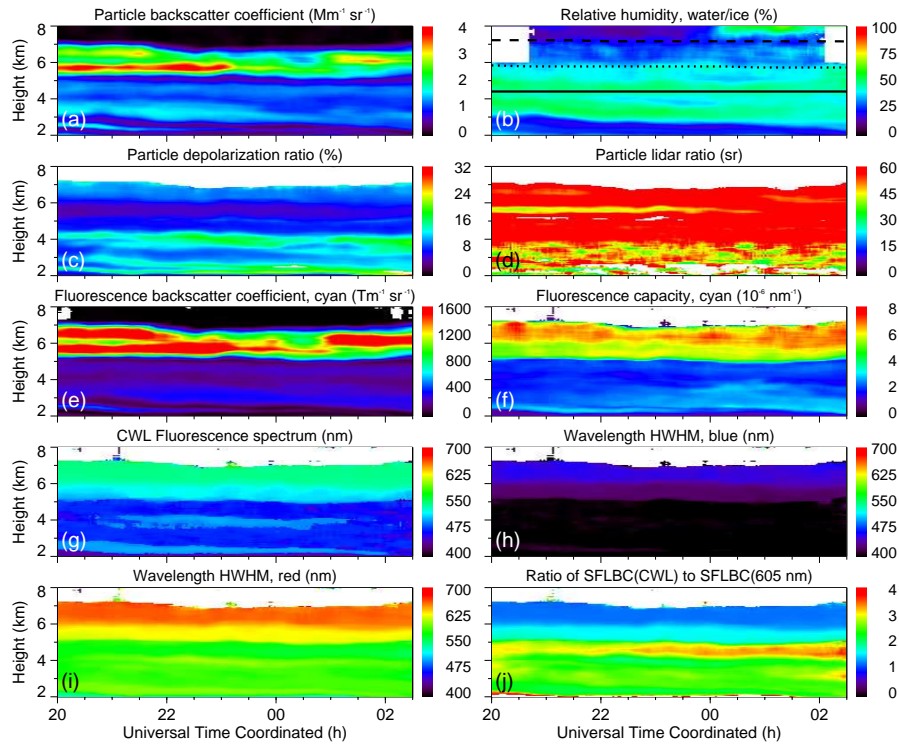

**Figure A4.** Similar to Fig. A1, but for the RAMSES measurement in the night of 19–20 July 2022 between 20:00 and 02:30 UTC. Black lines in panel **b** indicate 0 °C, -10 °C and -20 °C isotherms.



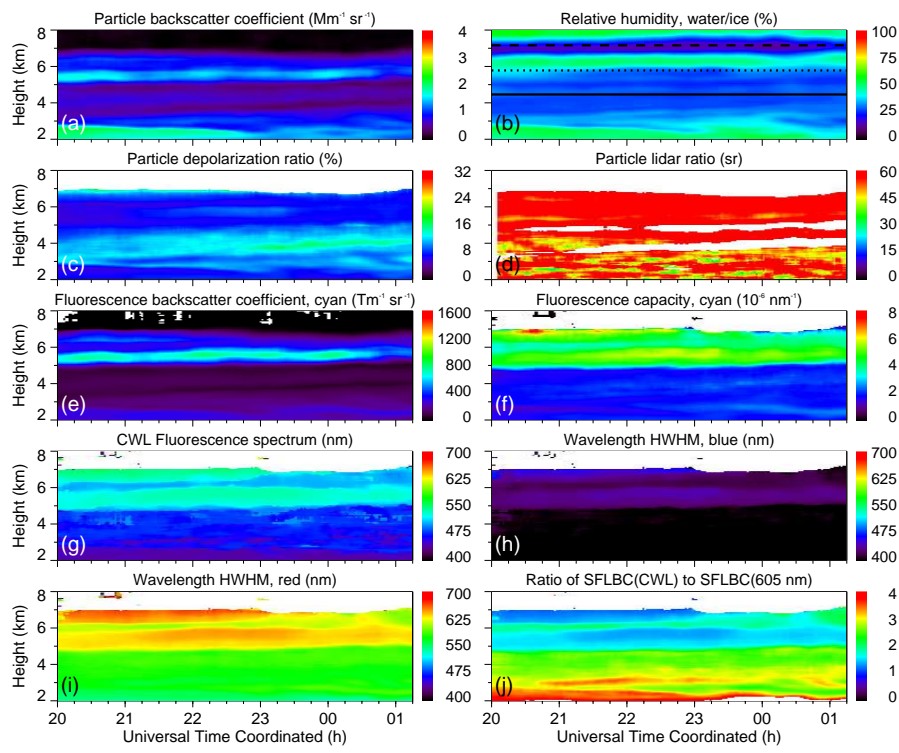

**Figure A5.** Similar to Fig. A1, but for the RAMSES measurement in the night of 20–21 July 2022 between 20:00 and 01:15 UTC. Black lines in panel **b** indicate 0 °C, -10 °C and -20 °C isotherms.





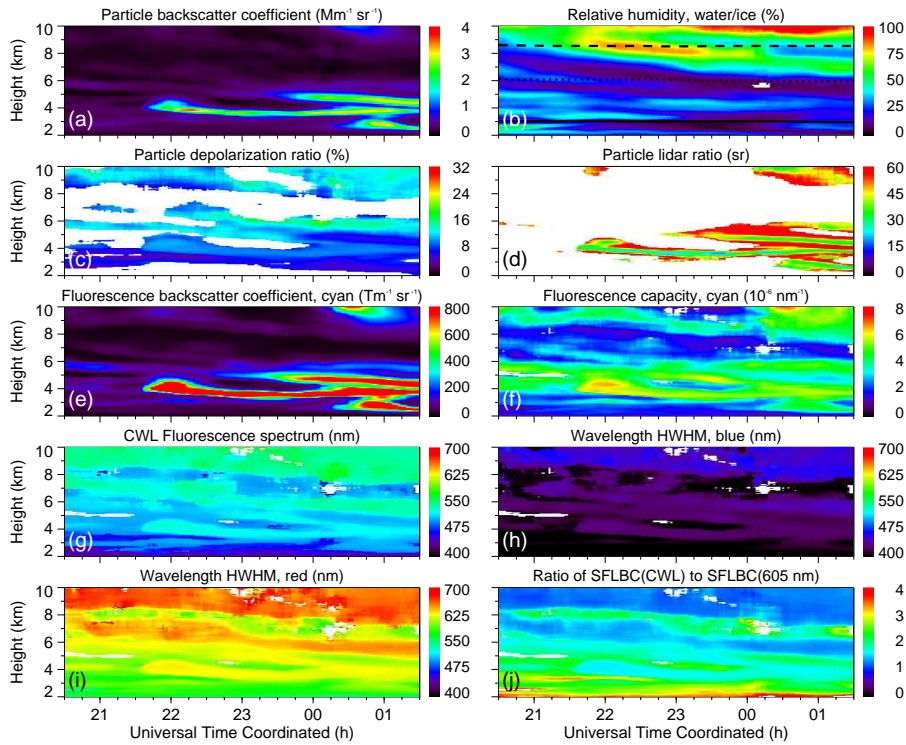

**Figure A6.** Similar to Fig. A1, but for the RAMSES measurement in the night of 1–2 June 2023 between 20:30 and 01:30 UTC. Black lines in panel **b** indicate 0 ˚C, -20 ˚C and -40 ˚C isotherms.



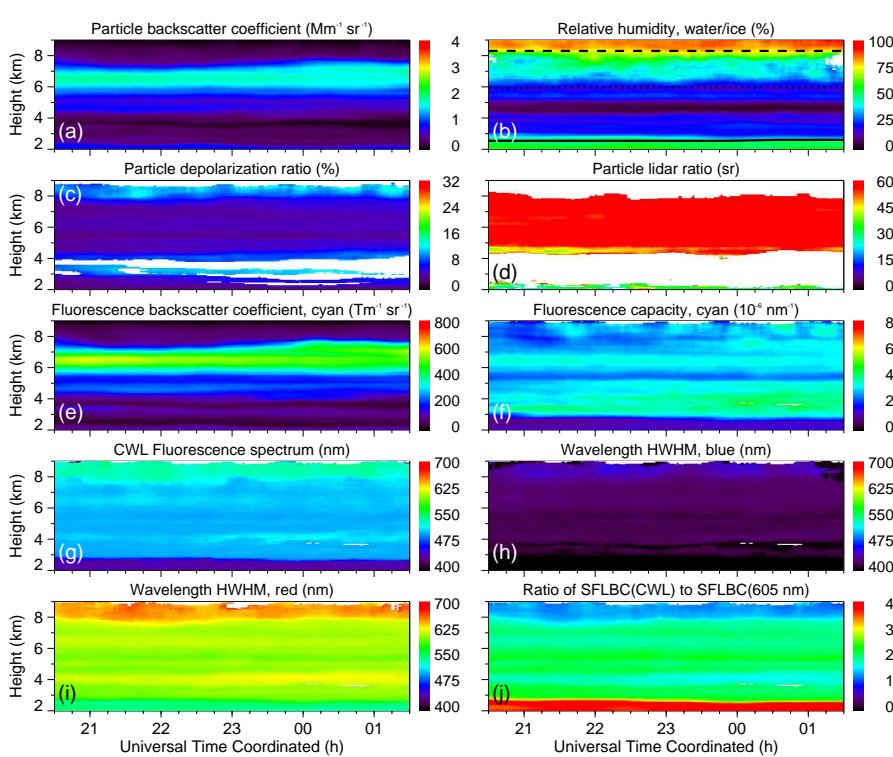

**Figure A7.** Similar to Fig. A1, but for the RAMSES measurement in the night of 4–5 June 2023 between 20:30 and 01:30 UTC. Black lines in panel **b** indicate 0 °C, -20 °C and -40 °C isotherms.



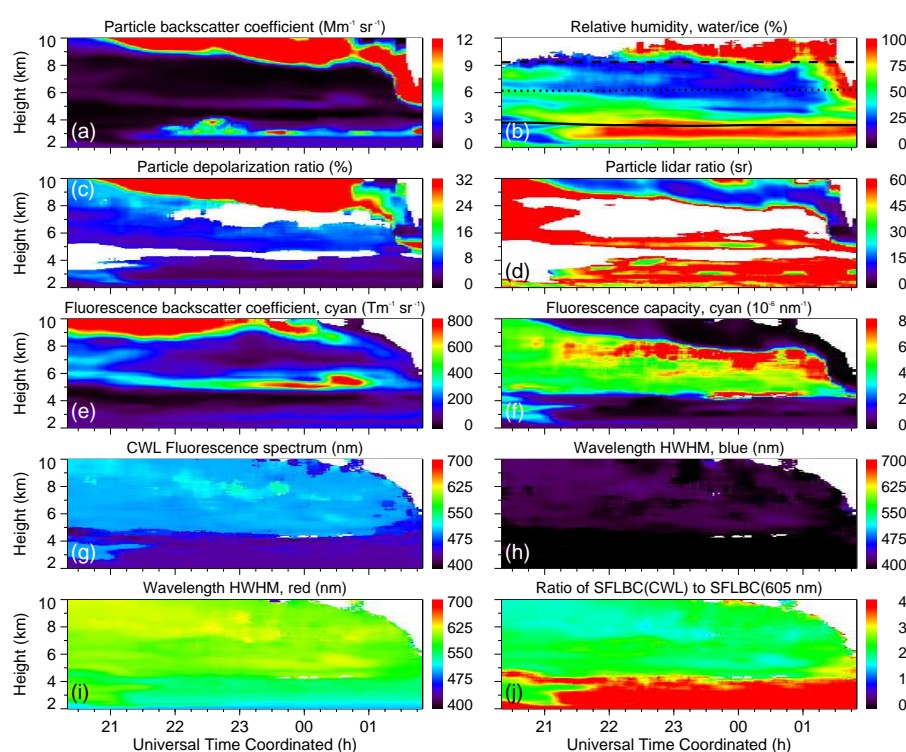

**Figure A8.** Similar to Fig. A1, but for the RAMSES measurement in the night of 9–10 July 2023 between 20:20 and 01:50 UTC. Black lines in panel **b** indicate 0 °C, -15 °C and -30 °C isotherms.



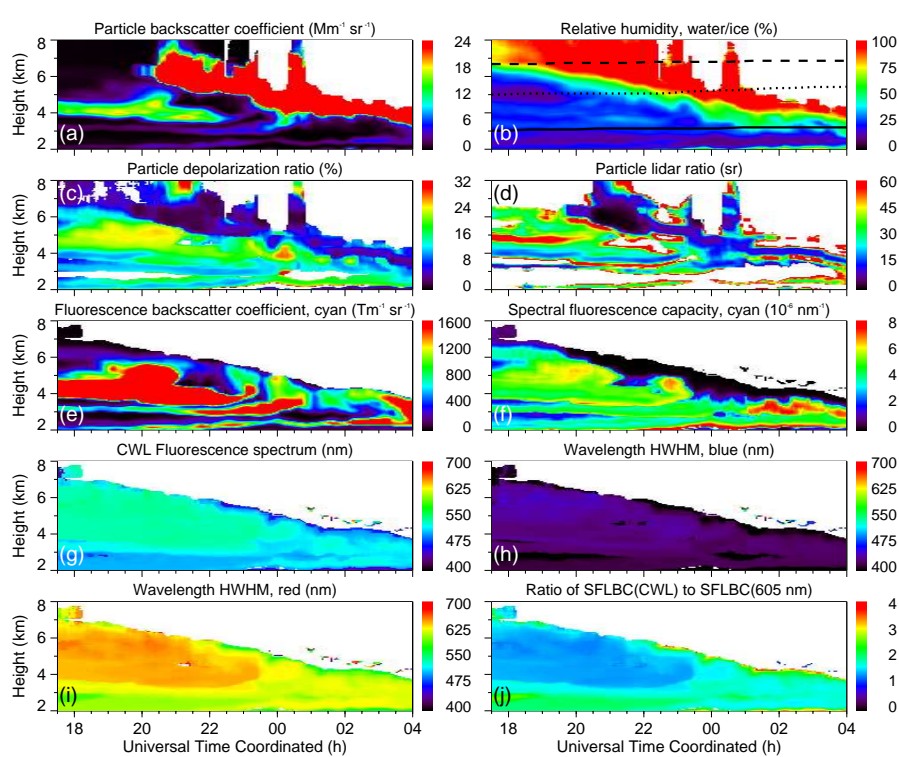

**Figure A9.** Similar to Fig. A1, but for the RAMSES measurement in the night of 30 September – 01 October 2023 between 17:30 and 04:00 UTC. Black lines in panel **b** indicate 0 °C, -10 °C and -20 °C isotherms.



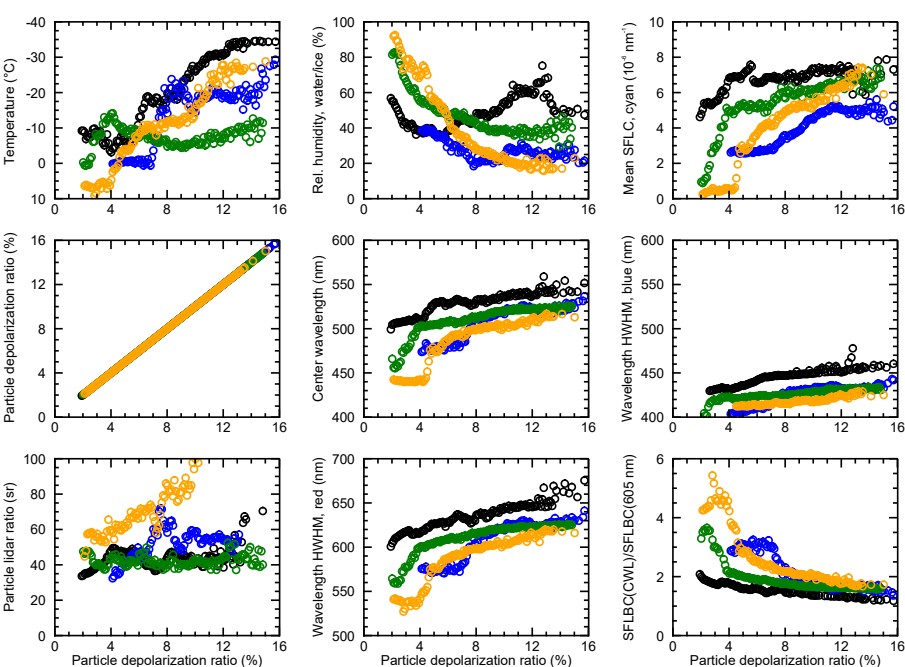

**Figure A10.** Dependence of selected atmospheric variables and elastic and inelastic aerosol properties on the particle depolarization ratio. Data points have a bin width of 0.1 % (absolute), mean values are shown. Measurement nights and data ranges are the same as in Fig. 15, results for 11–12 September 2020, 26–27 May 2023, 4–5 July 2023, and 9–10 July 2023 are illustrated with black, blue, green, and orange symbols, respectively.



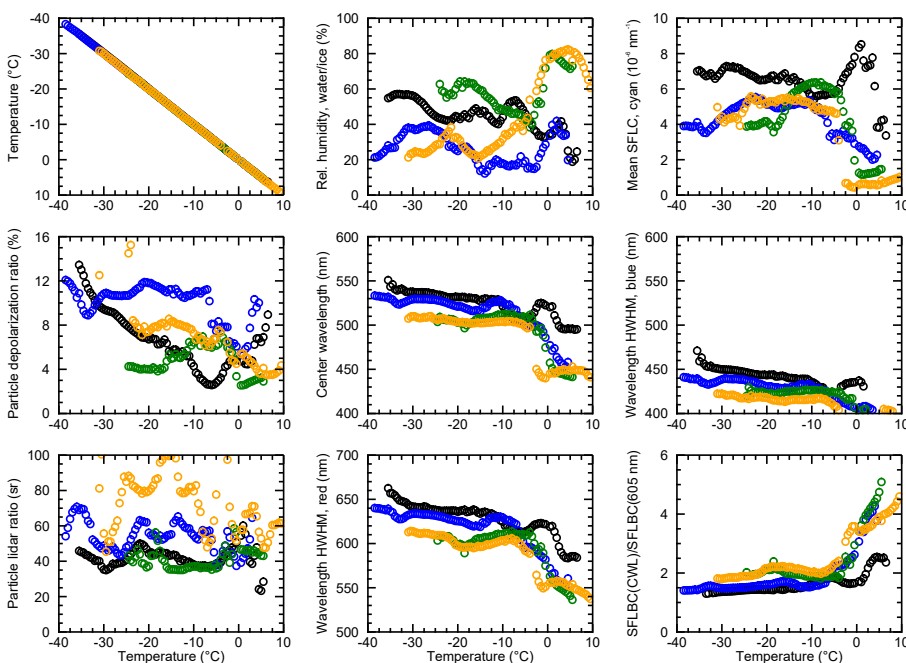

**Figure A11.** Similar to Fig. A10, but for ambient temperature as the independent variable. Data points have a bin width of 0.5 K, mean values are shown.