# Peer review of "Fluorescence spectra of atmospheric aerosols"

_EGUsphere, 2024_

## Author Response (AR1)

The authors thank the two anonymous reviewers for their positive assessment of our manuscript and their valuable comments and suggestions.

**Reviewer #1:**

Authors present results of high quality measurements of the fluorescence spectra in a wide range of height, up to the lower stratosphere. Main focus is done on the study of BBA and several important findings are presented, such as the red shift of fluorescence spectrum of BBA with height and blue shift in the clouds. The results presented are unique and are suitable for publishing in ACP. Manuscript is very well written and, in principle, can be published as it is.

I have just several minor comments.

1. The fluorescence lidar with several discreet channels can provide only rough estimate of the spectrum. Still red shift of BBA spectrum with height, as well as difference between spectra of BBA and the boundary layer aerosol was reported recently. So it should be cited.

The publication Veselovskii et al. (2025) has been cited.

2. Ln. 264. "In any case, at such low lidar ratios it can be ruled out that the BBA particles absorbed significantly."
But normally high lidar ratios are associated with strong absorption.

This is correct, high lidar ratios indicate strong absorption. Because the lidar ratios discussed here are < 40 sr, significant absorption can be excluded in this case.

3. Fig.14a presents temperature as a function of wavelength. I am confused. The plot can also be done for height (instead temperature), for example. Dependence will be probably similar. Why temperature is taken as a primary variable? The same in Fig.16: depolarization is plotted as function of central wavelength. Both parameters increase with height, but does it mean that parameters depend on each other?

Fig. 14a: We chose temperature as the primary variable because it is probably the transport feature that is best estimated by the HYSPLIT model. Indeed, temperature and height are well correlated, but in terms of fluorescence parameters, temperature has slightly higher correlation coefficients than trajectory height. It can also be assumed that fluorescence is more directly influenced by temperature than by altitude.
Fig. 16: These are only statistical correlations; a causal relationship has not been established. Yes, the central wavelength increases with altitude (which reflects the observed red shift of the fluorescence spectra), and yes, in general the depolarization ratio increases with altitude. However, this does not necessarily mean that depolarization ratio and fluorescence spectrum are causally linked, because different mechanisms could be responsible for the increases.

4. I am confused also with central plot in Fig.16: central wavelength vs central wavelength. What does it mean?

We wanted to relate a total of nine parameters to each other, three of them (center wavelength, depolarization ratio and temperature) as independent variables (see Figs. 16, A10 and A11). So that the arrangement of the panels does not change with each figure, we decided to always display all nine parameters as functions of the independent variable, accepting that one panel shows the identity.

5. Fig.20 "(center altitudes indicated in Fig. 19)". Would be more convenient to have altitudes at Fig.20 also.

The center altitudes have been included in the legend of panel c.

**Reviewer #2:**

General comment:

The results presented are of high quality and this study advances the understanding of connections between the laser-induced aerosol fluorescence spectrum and the type and origin of aerosol particles as well as atmospheric state variables. Therefore, this study is an important contribution to aerosol research and well-suited for ACP.

In general, the manuscript is well written, however, I would suggest minor revisions regarding the following aspects:

- Some figures, in particular those of the vertical profiles, depict a large number of different variables, most of which are only described in several lines of captions. This reduces readability and makes understanding more difficult. Please add more legends to the figures to improve readability. For details, please refer to the specific comments below.

- The manuscript is rather long and discusses several different aspects. Please consider to shorten the discussion at some points, which are not directly relevant for the scope of your study. For more details, please refer to the specific comments below.

Specific comments:

l. 1: "This study summarizes the results of 14 years of fluorescence measurements"
14 years seems a bit misleading here, as it gives the impression that you would present measurements over this whole time period. In fact, you exclusively present measurements from four years (2020-2023). Please consider to change the mentioned time period accordingly.

What is meant is that the analysis of the measurements presented does not come out of nowhere but is based on 10 years of previous groundwork which, incidentally, has also been used by other research groups sometimes without properly citing. The second paragraph of the introduction expresses this well with the sentence 'The present study is the essence of 14 years of fluorescence measurements with the spectrometric fluorescence and Raman lidar RAMSES at the Lindenberg Meteorological Observatory, Germany.', which is why we can accept the reviewer's suggestion here and reword the sentence.

ll. 156-7: "For an optimized visualization, values of aerosol index were set to −2, 0, and 10 if missing, < 0, and > 10, respectively."
This is not clear to me. When is the aerosol index set to these values? In Fig. 3, it seems that for none of the trajectories, values > 10 were estimated. Please clarify.

The value range for the plot has been narrowed down to make it clearer. For example, trajectories 05, 08, 14 and 22 had values of >10, which were set to 10.

Figure 2: Which meteorology data did you use for the trajectory analysis? Please add the coloring of the fire radiative power values visually to the legend. Maybe also indicate the starting dates for the trajectory analyses in both plots.

Data of the Global Data Assimilation System (GDAS) in 1° resolution were used. This information is now given in the running text.
Visualization is a matter of taste. We prefer not to overload the illustrations. The 3 colors of the radiant power are clearly distinguishable and are understandable because they are mentioned in the caption of Fig. 2. The start dates are shown in the captions; in our opinion, a repeated listing in the illustrations does not provide any additional information.

l. 174: Considering scenario 3 (weak fires as source region) seems only appropriate if the trajectory crossed these weak fires at mid or low altitudes. Did you consider the trajectory height for this?

The height of the trajectories was not included in the decision as to whether a trajectory is representative or not. The reason for this is that no clear limit height can be defined above which no BBA from weak fires can occur.

ll. 174-175: Under which criteria did you select the source regions from the representative S3 and S5 trajectories? For example, did you use weighting factors to account for the higher probability of S5 to be adequate?

All trajectories in categories S3 and S5 were considered equal candidates; no weighting factors were used to further sort the trajectories according to their probability. In the cases we examined, the source region could thus be narrowed down to a larger region (USA West, Canada West, Canada East, Russia, ...) in which all these trajectories crossed over fires. However, the trajectories often crossed several fires within this source region individually or one after the other, so that it was generally not possible to identify a single fire as the source of the aerosol. The mean for all trajectories of categories S3 and S5 was then used to calculate the transport duration and the average temperature along the trajectories.

Table 1: For better readability, I would suggest to add the descriptions of variables and abbreviations in the caption instead of placing them below the footnotes.

We would prefer this as well, but our understanding is that the style of the journal favors short table headings and other information should be given as footnotes. We will check with the layout editor.

Figure 4: Please adjust the distances between the left and right panel plots so that it becomes clear that the numbers belong to the color bars of the left panel plots and not to the y-axis of the right panel plots. Please consider to add the symbols of $\lambda c$, $\lambda b$ and $\lambda r$ to the respective plots for better readability.

We have increased the horizontal distance between the panels in all time-height plots. However, we do not consider it necessary to indicate the symbols for the wavelengths.

Figure 5: This is a lot of information and a bit confusing without visual guidance. Please add legends explaining the different quantities plotted to the figures. The same holds for Figs. 8, 11 and 19.
The blue and cyan curves in panel (e) are difficult to distinguish when close to each other. What is the averaging time of the profiles here?

The curves have been labeled in all profile plots. The colors cannot be changed because they stand for the corresponding color band. The integration time is 20 minutes for all profiles; this information can be found in the caption of Fig. 4 and now also of Fig. 5.

ll. 248-250: "only weakly developed features can be seen until about 01:00 UTC; later, in line with the reduction of $BFLcyan$, they shift to higher values and resemble those of the filament at 3.6 km"
This is not quite evident according to the figure. $\lambda r$ shows the main increase from 22 to 0 UTC already and seems to be the strongest increase, also $\lambda c$ increases from 23 UTC on.

This is of course a question of degree. Yes, variations can be seen in the variables mentioned, but in view of the dynamics that occurred in other measurement nights (see e.g. Fig. 10), these are comparatively small. By omitting the time information, the criticism is hopefully mitigated.

ll. 315-318: This is, in fact, an important finding. Could you provide evidence (maybe add the corresponding trajectory analysis in the Appendix) that the dust for the two time periods originated from different source regions?

We carried out a total of 19 HYSPLIT model runs at 5 different points in time, so it would go beyond the scope of this article to show this analysis in the Appendix. In order to take into account the even greater difficulties in determining the origin of the dust compared to BBA, we have formulated the text somewhat more cautiously.

l. 354: "changes" sounds like temporal variations here, but you describe differences between two aerosol layers. Thus, "differences" may be more suitable here.

The sentence has been reworded.

ll. 357-358: If you don't show the measurement from August 2017 and provide no reference, this statement is a bit out of context and can be omitted.

The sentence has been deleted. The measurement is as yet unpublished and therefore cannot be cited, and in view of the length of the manuscript it is not appropriate to show it in the appendix. However, because it is so interesting, it should be briefly discussed here. The measurement was taken with RAMSES during the night of 30-31 August 2017. The figure shows night averages. The fluorescence backscattering coefficient was measured at 407.5 nm with the detection channel normally used for water vapor measurement. The spectral fluorescence capacity is $2.9 \times 10^{-6}$ nm$^{-1}$, which fits well with a red-shifted BBA fluorescence spectrum. The comparatively high particle depolarization ratios and low lidar ratios suggest that the BBA particles were irregular, of significant size and only slightly absorbing.

[Figure]

Figure 11: See comment to Fig. 5. Especially if the caption refers to another figure, legends should be present, explaining the different quantities that are shown.

See reply to comment to Fig 5.

l. 376: What do you mean with atmospheric state here? The ambient conditions during the transport of the aerosol particles or the state of the atmosphere during the measurement. Please clarify.

Both could be true, but the latter is meant here.

ll. 384-400: As you state yourself at the beginning of the paragraph, this excursion to the boundary layer is not in the focus of your study. Furthermore, the situation is complex and conclusions are difficult from the two presented cases only. Please consider omitting this paragraph to shorten the long section a bit and keep focused on the main scope of the manuscript (studies of the free troposphere).

Even though the focus is on BBA measurements in the free troposphere, we think it is important and instructive to compare them to those of other aerosols and at other altitudes. For example, fluorescence spectra are shown that are typical for aerosols in the boundary layer or for Saharan dust in the free troposphere. We therefore also consider it appropriate to discuss the anomaly of BBA in the boundary layer, which does not occur frequently.

ll. 411-412: Please clarify how you understand the "ensemble-averaged mean temperature". Is it the mean temperature of the air mass during transport averaged over the representative trajectories?

Yes, that is correct.

ll. 490-492: "The same also largely applies to T, which is anti-correlated with $\lambda c$ (once again demonstrating the general red shift of the BBA fluorescence spectrum with height), and to $\delta$ par, albeit more weakly."
For $\delta$par, this is not evident according to the figure. There seem to be differences between the different measurement nights.

That is certainly correct. For particle depolarization ratio, the trend is clearly recognizable in all measurement nights, but the trend lines are not as close together as with temperature or capacity. And so it is again a question of degree as to how many deviations one accepts in the assessment. The reviewer is critical in this case, we follow his judgment and delete the corresponding text.

l. 548: The fluorescence backscatter coefficients decline with the **increase in** particle backscatter coefficient, don't they?

Yes, thanks. Corrected.

ll. 556-557: As you have already cited in the summary, this finding has recently been reported by Gast et al. (2024). Please add the reference also here in the main text.

The publication Gast et al. (2024) has been cited.

Figure 20: Please also add the center altitudes of the displayed layers in the figure for better readability.

The center altitudes have been included in the legend of panel c.

l. 643: see comment to l. 1

See reply to l. 1.

Typos:

l. 411: In Fig. 14a, (add a comma).
l. 412: in Fig. 14b, (add a comma).
ll. 568-569: According, to the particle backscatter coefficient, (shift the comma)
l. 643: fluorescence measurements of aerosols

All typos have been corrected.